# Holes in Latent Space: Topological Signatures Under Adversarial Influence

## Abstract

Internal model representations are central to driving interpretability in machine learning and understanding them is key to reliability. Using persistent homology (PH), a technique from topological data analysis that captures the shape and structure of data at multiple scales, we present a global and local characterization of the latent space of three state-of-the-art Large Language Models (LLMs) under two adversarial conditions. Through a layer-wise topological analysis, we show that adversarial interventions consistently compress the latent space, reducing topological diversity at smaller scales while amplifying prominent structures at higher scales. Critically, these topological signatures are statistically meaningful and remain consistent across model architectures and sizes. We further introduce a novel neuron-level interpretability framework where PH is used to quantify information flow within and across layers. Our results establish PH as a powerful tool for interpretability in LLMs and for detecting distinct operational modes under adversarial influence.

## 1. Introduction

Understanding the behavior of large language models (LLMs) is an active area of machine learning research and is crucial for ensuring their reliability, transparency, and fairness; it enables the identification of biases, vulnerabilities to adversarial inputs, and the potential for unintended consequences in real-world applications. Examining their latent space offers a promising way to interpret their internal mechanisms, including feature representations (Cunningham et al., 2023), task structures (Hendel et al., 2023), and decision boundaries (Zhao et al., 2024). Latent space analysis has also been used to control model behavior across

various linguistic and Natural Language Processing (NLP) tasks, such as style transfer (Turner et al., 2024) and domain adaptation (Subramani et al., 2022), as well as on detecting adversarial threats (MacDiarmid et al., 2024; Abdelnabi et al., 2024; Zou et al., 2024).

Most existing studies on the latent space focus on linear structures, overlooking the nonlinear and topological transformations that high-dimensional activation spaces might undergo (Brüel-Gabrielsson et al., 2020; Kirch et al., 2024). As a result, little is known about how adversarial manipulations reshape model representations or whether these effects generalize across architectures and threat scenarios.

In this paper, we address this gap using persistent homology (PH)—a technique from topological data analysis (TDA) that captures the *shape* and *size* of data and encodes this information in *barcodes* that represent the span of multi-scale topological features in the data. A motivating example of the clear effectiveness of PH in distinguishing between normal and adversarial activations is given in Figure 1. In this paper, we conduct an extensive study of the topology of LLM representation spaces and its implication in distinguishing normal and adversarial representations.

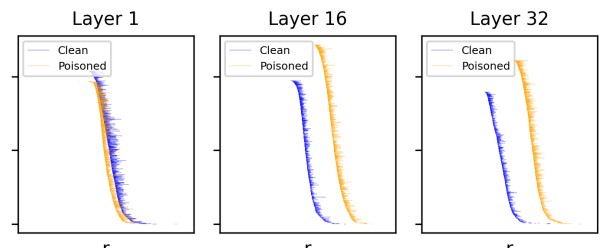

*Figure 1.* **Example barcodes from clean vs. poisoned activations.** Barcodes in dimension 1 for two samples of $k = 1000$ activations of clean (blue) and poisoned (orange) activations of Mistral 7B at layers 1, 16 and 32.

Our key results can be summarized as follows.

- We conduct a **layer-wise global PH analysis** across multiple model architectures and sizes and achieve a near-perfect separation of normal vs. adversarial activations, demonstrating that PH robustly captures the structural deformations induced by adversarial influence.

[1]Anonymous Institution, Anonymous City, Anonymous Region, Anonymous Country. Correspondence to: Anonymous Author <anon.email@domain.com>.

Preliminary work. Under review by the International Conference on Machine Learning (ICML). Do not distribute.

- We provide insights into the **effect of adversarial attacks on the representation space**. Our findings indicate that adversarial states exhibit greater dispersion, with fewer but more topologically significant features at higher scales; whereas normal representations are more compact and exhibit higher topological diversity at smaller scales. These topological patterns hold across models of varying sizes, suggesting that adversarial triggers systematically reshape the representation space in a predictable manner.

- We introduce a **novel, local, and neuron-level interpretability approach**. By mapping neuron activations from pairs of consecutive and non-consecutive layers into 2D coordinates and applying PH, we capture fine-grained structural changes and information flow within the network. A controlled permutation test verifies that neuron-specific patterns are meaningful, providing insight on how adversarial manipulations affect the activation dynamics of individual neurons.

### 1.1. Related Work

TDA methodology coupled with machine learning has been an active approach to analyzing text data in recent years; see Uchendu & Le (2024) for a comprehensive survey of TDA applications in NLP.

**TDA and Language Model Representations.** While TDA has shown promise in analyzing LLM representations (Chauhan & Kaul, 2022; García-Castellanos et al., 2024), its application to understanding misaligned behavior is largely unexplored. Existing work primarily focuses on single-model analysis, such as the evolution of topological features across layers in BERT (Chauhan & Kaul, 2022) or RoBERTa (García-Castellanos et al., 2024). Recently, Gardinazzi et al. (2024) introduced a persistence similarity metric to find consistent topological features across different LLMs, layers, and hyperparameters.

**Geometric Properties and Adversarial Detection using LLM Representations.** LLM representations exhibit structured geometric properties, which interpretability research uses to identify linear relationships between semantic and factual attributes (Vaswani et al., 2023; Nanda et al., 2023; Marks & Tegmark, 2024; Gurnee & Tegmark, 2024). Sparse autoencoders extract non-orthogonal features (Cunningham et al., 2023; Li et al., 2024b), yet most analyses rely on linear projections, potentially overlooking stable higher-order geometric structures (Brüel-Gabrielsson et al., 2020). These limitations become apparent in adversarial and safety contexts, where detecting various attack types often relies on linear probes or shallow classifiers (Chao et al., 2023; CH-Wang et al., 2024; Abdelnabi et al., 2024).

## 2. Background

In this section we present the TDA methodology used in this study as well as details on the adversarial influences we study in LLM representations.

### 2.1. Persistent Homology

Persistent homology is a core technique in TDA for extracting multi-scale structural features (e.g., connected components, loops, and higher-dimensional voids) from complex datasets (Carlsson, 2009). It adapts the mathematical concept of homology from pure algebraic topology to data taking the form of finite metric spaces using a nested sequence of topological spaces known as a *filtration*, and outputs a compact summary of the topological features known as a *barcode*.

In this paper, we consider simplicial complexes as our topological spaces, due to the availability of tractable algorithms for computing simplicial homology. More specifically, we study Vietoris–Rips simplicial complexes (Vietoris, 1927) at scale $r \in [0, \infty)$, denoted by $K_r$, and defined as the family of all simplices that can be formed from the set of vertices given by $S$ of diameter less than or equal to $r$. A filtration is a parameterized set of simplicial complexes $\{K_r : r \in R\}$, where $R$ denotes a totally ordered indexing set and $r \leq s$ implies $K_r \subseteq K_s$; the *Vietoris–Rips filtration* is the family of Vietoris–Rips complexes as $r$ evolves over $[0, \infty)$.

The *persistence barcode* is a collection of intervals summarizing the lifetimes of topological features that are born, evolve, and die as the filtration parameter evolves; each bar corresponds to a distinct topological feature with its starting/end point corresponding to its birth/death time. Vietoris–Rips PH is the construction of choice in many applications due to the fast computational software RIPSER (Bauer, 2021); see Figure 2 for an example of a Vietoris–Rips filtration and its corresponding barcode. For complete details on the general theory of PH, see Zomorodian & Carlsson (2005) and Oudot (2017).

To compute PH of LLM representations, we can represent all hidden-layer representations as a 3D tensor $\mathbf{X} \in \mathbb{R}^{N \times L \times D}$, where $N$ is the number of input examples, $L$ is the number of layers, and $D$ is the dimension of each hidden representation vector. For each layer $\ell \in \{1, \ldots, L\}$ and input example $n \in \{1, \ldots, N\}$, the hidden-layer activation $\mathbf{x}_{n,\ell} \in \mathbb{R}^D$ is then a point in a $D$-dimensional space. The activations then make up a point cloud on which we compute PH.

### 2.2. Adversarial Influence on LLMs

We aim to characterize the topological distinctions in a LLM's representation space when it operates under *normal* versus *adversarial* modes of behavior. To this end, we exam-

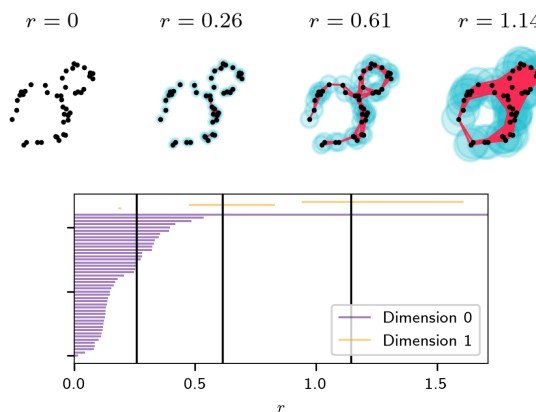

*Figure 2.* **Top:** Vietoris–Rips filtration constructed from a sample of 45 points over 2 circles with noise at four values of the filtration parameter or distance threshold $r \in [0, \infty)$. **Bottom:** corresponding persistence barcode for the 0- and 1-dimensional topological features (components and loops, respectively), with vertical lines corresponding to the filtrations values displayed above.

ine two distinct attack scenarios—each with its own normal and adversarial class—which we introduce as follows.

**Extended Prompt Injection (XPIA).** In XPIA, an attacker embeds hidden or malicious instructions (often via retrieval-augmented content like websites and documents) into a user's legitimate prompt (Greshake et al., 2023). These injected instructions can override the model's default behavior to produce harmful output and exfiltrate data (Yi et al., 2023). In this situation, we denote the normal mode as *clean* and the adversarial mode as *poisoned*.

**Backdoor Fine-Tuning and Sandbagging.** By contrast, backdoor fine-tuning modifies the LLM to behave differently when it receives a "trigger" phrase (Hubinger et al., 2024; Greenblatt et al., 2024). A special case, *sandbagging*, involves intentionally suppressing the model's observable capabilities until a hidden key phrase is provided, at which point it *unlocks* its full performance (van der Weij et al., 2024). We specifically focus on a developer-driven variant of sandbagging, where the model's underperformance is deliberately orchestrated during fine-tuning. We define the normal mode as *fully elicited*, where the password (trigger) is present and the model operates at full capacity, and the adversarial mode as *locked*, in which the password is absent and the model deliberately underperforms.

## 3. Method and Experimental Setup

In this section, we give details on the data we analyze and outline the computational pipelines we implement for the global and local analyses. More details on the latter can be found in Appendix B.1 and Appendix C.1.

### 3.1. Data and Representations

We study the above two adversarial scenarios—(*i*) XPIA and (*ii*) sandbagging—by analyzing the *representation vectors* of the last token in the input prompt which encodes the model's aggregated interpretation of the input context (Zou et al., 2023). Let $\mathrm{Hidden}_\ell(x)[-1] \in \mathbb{R}^d$ denote the layer-$\ell$ representation of the last token from the input prompt $x$.

**TaskTracker (Clean vs. Poisoned).** We leverage the TASK-TRACKER *test* set ($> 62k$ examples), which draws primary tasks or user prompts from the SEP (Zverev et al., 2024) and HotPot QA (Yang et al., 2018) datasets. Each example pairs a primary task with one or more data blocks that may be *clean* or *poisoned*. Poisoned blocks include diverse payloads with injected tasks spanning code generation (Chaudhary, 2023)) and malicious and jailbreak instructions across several attack datasets. Dataset description and its confounds minimization (e.g., domain shifts, prompt length, payload position) is in Appendix B of Abdelnabi et al. (2024).

For each test example, TASKTRACKER provides the layer-$\ell$ representation of the last token from the input prompt $x$ for the primary task $\mathrm{Act}_\ell^{(\mathrm{pri})}(x) = \mathrm{Hidden}_\ell(x_{\mathrm{primary}})[-1]$, and for the task and data block $\mathrm{Act}_\ell^{(\mathrm{full})}(x) = \mathrm{Hidden}_\ell(x_{\mathrm{primary+data}})[-1]$. Throughout our experiments, we leverage their difference, $\Delta\mathrm{Act}_\ell(x) = \mathrm{Act}_\ell^{(\mathrm{full})}(x) - \mathrm{Act}_\ell^{(\mathrm{pri})}(x)$, which quantifies how the data block alters the final-token representation. We study the topological features of this representation data from Phi3 3.8 (Abdin et al., 2024), Mistral 7B (Jiang et al., 2023), and LLaMA 8B (Grattafiori et al., 2024).

**Sandbagged Models (Locked vs. Elicited).** To study *sandbagging*, we fine-tuned Mistral 7B and LLaMA3 8B on 3,347 synthetic WMDP-like questions (inspired by Li et al. (2024a)) using LoRA (Hu et al., 2021), following the framework of van der Weij et al. (2024). We extracted final-token representations from each layer when prompted with evaluation prompts, both with (elicited) and without (locked) the password. See Appendix D for further details.

### 3.2. Global Layer-Wise Analysis: Descriptive and Inferential Methods on Barcode Summaries

This analysis aims to highlight the intrinsic differences in the topology of normal versus adversarial activations, facilitating interpretations for the underlying causes of these distinctions. To this end, we implement the pipeline in Figure 3. We use RIPSER (Bauer, 2021) to compute barcodes based on Vietoris–Rips filtrations, leveraging subsampling techniques (e.g., Chazal et al. (2015)) to mitigate the computational constraints of PH, which make it impossible to compute the barcode of any of our whole datasets. Following Ali et al. (2023), we vectorize these barcodes into 41-dimensional *barcode summaries*—more details on the

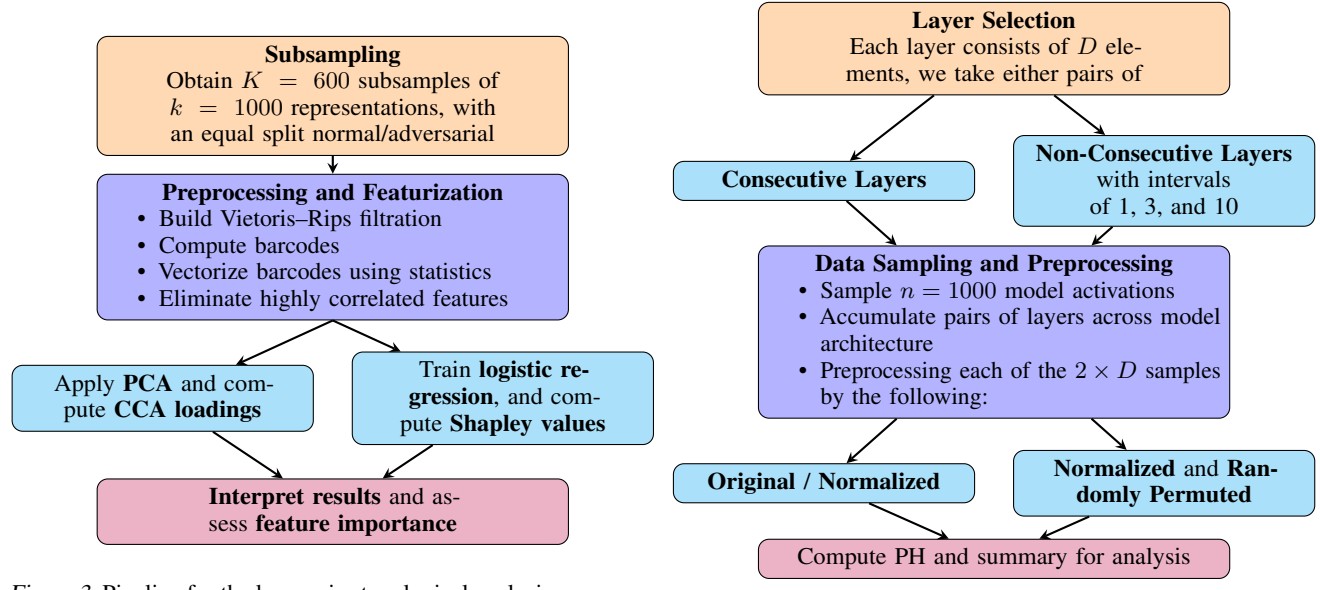

*Figure 3.* Pipeline for the layer-wise topological analysis.

*Figure 4.* Pipeline for local analysis

exact statistics computed can be found in Section 4.1 and Appendix B.1. To refine the feature set, we apply cross-correlation analysis to remove highly correlated variables, ensuring an efficient and informative representation. We perform a PCA and compute canonical correlation analysis (CCA) loadings to investigate feature importance. We also train a logistic regression and compute Shapley values (Lipovetsky & Conklin, 2001) to evaluate the predictive power of features.

### 3.3. Local Information Flow Analysis: Study of Consecutive and Non-Consecutive Layers

We study the information flow between consecutive and non-consecutive layers by analyzing element-wise interactions within activation spaces. Instead of aggregated representations, our interpretability strategy focuses on neuron-level activations, mapping the weights of individual neurons in consecutive layers into a 2D coordinate space. We then apply PH to uncover meaningful structural patterns in these localized activations, providing granular insights into network behavior and the role of specific neurons in shaping model representations. We used control condition to validate that neuron activations capture meaningful signals, testing if randomly shuffled neuron indices in one activation vector preserves or perturbs topological patterns. If neuron-specific weights encode valuable information, shuffling will distort these patterns. We extend this analysis to non-consecutive layers, examining neuron interactions across intervals of 1, 3, and 10 layers. If adjacent layers act on similar neuron groups, we expect to identify distinct topological features; otherwise, trends should resemble those observed under random shuffling. Figure 4 illustrates the pipeline for such analysis; further details appear in Appendix C.1.

## 4. Results

We present the results and associated statistical conclusions from our global and local study in this section.

### 4.1. Global Results

Figure 3 illustrates the results of the Mistral 7B analysis for the clean vs. poisoned dataset, where barcodes are computed using the Euclidean distance between activations. Similar results for LLaMA3 8B and Phi3 can be found in Appendix B.2, as well as results computing the barcodes using the cosine distance in the activation space. We present analogous results for locked vs. elicited in Appendix B.3, highlighting differences in correlation patterns and the interpretability of feature influence, which are less clear.

**Cross-Correlation Analysis of Barcode Summaries.** We study the cross-correlation matrix of the 41-dimensional barcode summaries obtained from the subsamples. These include 35 statistics derived from a $7 \times 5$ grid of {mean, minimum, first quartile, median, third quartile, maximum, standard deviation} $\times$ {death of 0-bars, birth of 1-bars, death of 1-bars, persistence of 1-bars, ratio birth/death of 1-bars}; as well as the total persistence (i.e., sum of the lengths of all bars in the barcode), number of bars, and persistent entropy (Chintakunta et al., 2015; Rucco et al., 2016) for 0- and 1-bars (see Appendix A.1 for a precise definition).

The results in Figure 5 show that a growing block of highly correlated features appears in the layers of the model. Setting a threshold of correlation equal to 0.9, this block includes all statistics except for the minimum, maximum, and standard deviation of the deaths of 0-bars, the mean, median,

and third quartile of the births and deaths of 1-bars and the total persistence of 0-bars; and all statistics for births and deaths of 0- and 1-bars except for the minimum of the deaths of the 0-bars, the mean, median, third quartile and standard deviation of the persistence of 0-bars for layers 16 and 32.

The mean death of 0-bars emerges as the first prominent feature, so we retain it as our representative of topological features in our analyses. We remark here that the prominence of statistics related to dimension 0 persistent homology, which corresponds to connected components rather than higher-dimensional holes, does not imply a lack of significance for higher-order homology (specifically, dimension 1). Here, we observe a strong correlation between dimension 0 and dimension 1 statistics and mathematically, it is known that the deaths of 0-bars are closely linked to the births of 1-bars—this relationship has been explored in the context of Morse theory; see Adler & Taylor (2011) for further discussion. Thus the mean death of 0-bars inherently captures information on 1-bars as well.

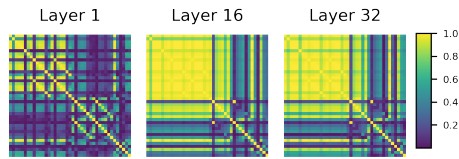

*Figure 5.* **Cross-correlation matrices for barcode summaries of clean vs. poisoned activations.** Growing block of correlated features appears in the cross-correlation matrix of the barcode summaries.

In light of the cross-correlation results, we discard all features that have a correlation higher than a threshold of 0.5 with at least one feature present in the analysis, admitting a few more features in the blocks described above. The resulting data set will be called the *pruned barcode summaries*, see Table 1 for details.

**PCA and CCA.** Figure 6 presents PCA results of the pruned barcode summaries, showing a clear separation between clean and poisoned subsamples across layers, consistent with the motivating experiment on a single subsample of clean vs. poisoned activations (Figure 1). We now test the impact of each individual feature on the appearance of this separation.

The projection of features onto the first principal component in Figure 6 reveals that, for layers 8 and 16, the mean of the deaths of 0-bars is the dominant contributor, while it is the second most significant for other layers. In layer 0, the primary contributor is the standard deviation of 0-bar deaths, in layer 23, the number of 1-bars, and in layer 32, the mean birth-to-death ratio of 1-bars. We support this observation with a CCA between the pruned barcode summaries and the principal components of the PCA. CCA is a statistical

*Table 1.* **Pruned barcode summaries for layers 1, 16, and 32.** Features from the barcode summaries with correlation less than 0.5 in the cross-correlation matrix.

|  | Layer 1 | Layer 16 | Layer 32 |
|---|---|---|---|
| Mean death 0-bars | ✓ | ✓ | ✓ |
| Maximum death 0-bars | ✓ |  |  |
| Standard deviation death 0-bars | ✓ |  |  |
| Minimum birth 1-bars | ✓ |  |  |
| Maximum birth 1-bars | ✓ |  |  |
| Minimum persistence 1-bars | ✓ | ✓ | ✓ |
| Maximum persistence 1-bars | ✓ |  |  |
| Mean birth/death 1-bars |  | ✓ | ✓ |
| Maximum birth/death 1-bars |  | ✓ | ✓ |
| Total persistence 1-bars |  |  | ✓ |
| Number 0-bars | ✓ | ✓ | ✓ |
| Total features | 8 | 5 | 6 |

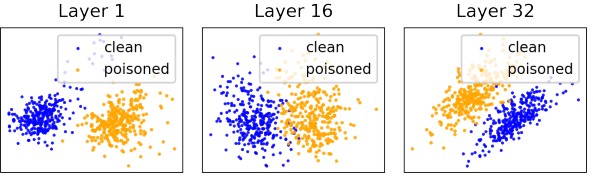

*Figure 6.* **PCA of barcode summaries of clean vs. poisoned activations.** Clear distinction appears in the projection onto the two first principal components from the PCA of the pruned barcode summaries of *Mistral_7B* for layers 1, 16, and 32.

method that quantifies linear relationships between two multivariate datasets by finding pairs of canonical variables with maximal correlation. The *loadings* are the contributions of individual features to these canonical variables, measuring their importance in capturing the relationship. The five largest loadings per layer can be found in Figure 18. We again find that mean of the deaths of the 0-bars ranks first in all layers, with a significantly higher magnitude compared to the remaining contributions. The second most prominent contributor is the standard deviation of the deaths of 0-bars for layer 1 and the mean ratio birth/death for 1-bars for layers 16 and 32.

**Regression and SHAP Analysis.** We train a logistic regression on the pruned barcode summaries, with a 70/30 split between train and test. The results of the regression plotted in the PCA projection, for visualization purposes, can be found in Figure 7. We obtain perfect accuracy and AUC–ROC, when testing on the test data, and 5-fold cross validation over the training data for all models.

We use Shapley (or SHAP) values to interpret the exceptional performance of the regression model. Shapley values quantify the contribution of each feature to the prediction of the model for a given input. Figure 8 shows beeswarm

plots of Shapley values for layers 1, 16, and 32, where each row represents a feature, and points correspond to the SHAP values of the input data (spread across the $x$-axis), colored by feature value for the corresponding input data point. The analysis reveals that the mean of 0-bar deaths strongly influences predictions, exhibiting a clear dichotomous effect.

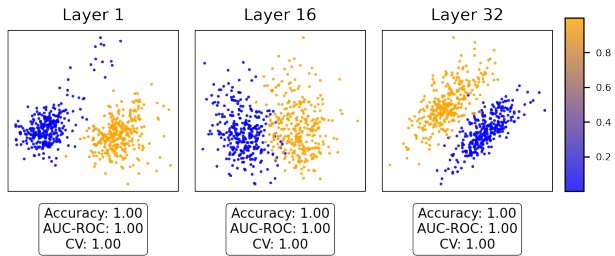

Accuracy: 1.00
AUC-ROC: 1.00
CV: 1.00

Accuracy: 1.00
AUC-ROC: 1.00
CV: 1.00

Accuracy: 1.00
AUC-ROC: 1.00
CV: 1.00

*Figure 7.* **Logistic regression for clean vs. poisoned activations.** Prediction of a logistic regression trained on a 70/30 train/test split of the pruned barcode summaries, plotted on the projection onto the two first principal components for visualization purposes.

### 4.2. Interpretation: The Shape of Adversarial Influence

Interpreting the distributions of the barcode summaries for clean vs. poisoned data reveals that adversarial conditions typically yield fewer dimension-1 loops forming at later scales, yet persisting longer (see Figure 9). Conversely, the non-adversarial conditions tend to form earlier loops with more uniform lifetimes (higher persistent entropy). This is also foreshadowed in the results of the Shapley values (Figure 8) which exhibit that lower values for the mean of the death of 0-bars generally shift predictions toward 0, i.e., "clean"; while larger values classify inputs as "poisoned."

In addition, a local dispersion ratio (Appendix A.2) and average cosine distance (cf. Figure A.3) substantiate these results, revealing that adversarially influenced representation vectors become more dispersed (higher cosine distance) or concentrate variance onto specific axes (leading to flips in dispersion ratio), implying a reallocation of representational capacity toward a smaller number of large-scale features. In contrast, the non-adversarial conditions produce lower or more stable distance measures (less reconfiguration in the hidden space).

Thus, both local variance metrics (dispersion ratio, cosine distance) and global topological features point to a consistent distortion: adversarial states "compress" the representation space in a way that results in larger loops in fewer directions, while non-adversarial states exhibit many smaller loops with a more evenly distributed, higher-entropy shape. See Appendix A.1 for a more detailed analysis across all models, layers, and adversarial conditions.

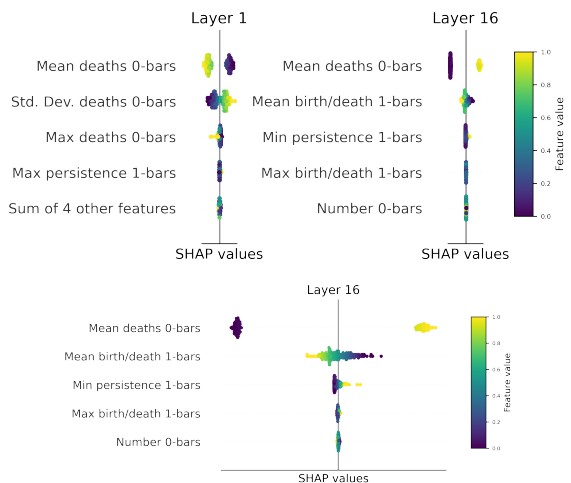

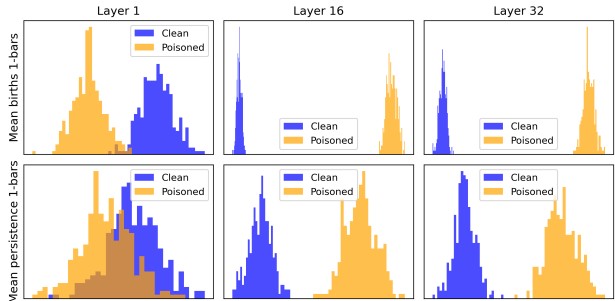

*Figure 8.* **SHAP analysis: clean vs. poisoned activations.** Beeswarm plot of logistic regression SHAP values trained on the pruned barcode summaries for layer 1, 8, 16, 24, and 32.

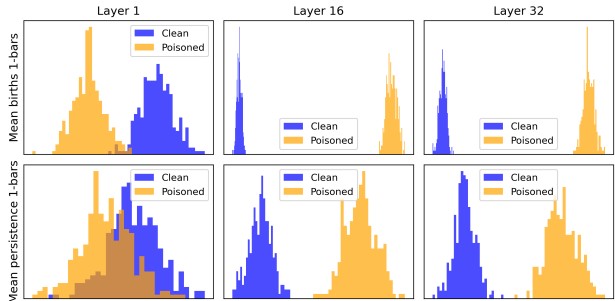

*Figure 9.* **Histograms for the mean of the births of 1-bars and persistence of 1-bars for Mistral.** Features extracted from the barcode summaries of the activations for layers 1, 8, and 32 of the clean vs. poisoned dataset.

### 4.3. Local Results

We present the results of the local analysis following the pipeline described in Section 3.3 on the Mistral model; similar findings for the Phi3 and LLaMA3 models can be found in Appendix C.2.

**Consecutive Layers Analysis.** We investigate the element-wise interactions within consecutive pairwise layer representations derived from the neural network using the Vietoris–Rips filtration. To quantify these interactions, we compute a selection of barcode summaries—such as total persistence and the mean birth or death times for 0- and 1-bars—for both clean and poisoned samples. To verify that these features do not merely arise due to differences in scale between clean and poisoned activations, we also compute the same barcode summaries after rescaling the activation values to have zero mean and unit variance.

The top left and middle plots of Figure 10 compare the average total persistence of 1-bars between clean and poisoned activations, without and under scaling. These plots reveal variations in the total persistence of poisoned activations relative to clean ones, though these differences appear somewhat reduced due to scale. In the top right plot of Figure 10, we present the difference in total persistence of 1-bars between clean and poisoned activations under a "control" condition. This control applies both scaling and random permutation to disrupt element-wise correspondences between layers, mimicking a neural network without local interactions, thereby suppressing meaningful topological features. Accounting for differences in scale, we find that difference in total persistence under this control setting is significantly smaller than in the original and scaled activations.

To examine the relationship between barcode summaries of clean and poisoned activations more finely, we compute their ratio, shown in the bottom left plot of Figure 10. Notably, for the original (raw) activations, the ratio exhibits a decreasing trend, crossing the value of 1 around layer 12. This shift reflects a transition in relative behavior: initially, total persistence is greater for clean activations, but beyond this point, it becomes greater for poisoned activations. This may suggest that, initially, interactions within the poisoned activations were more constrained. However, as the model learns across layers, this constraint gradually shifts and is instead applied to the clean activations. In comparison, the ratio for scaled activations is more subtle, showing no consistent trend across layers. However, when comparing this ratio to that of the control setting (scaling and permutation), which remains closely aligned with the baseline ratio = 1, we observe a distinct deviation, confirming that the observed behavior is not merely an artifact of scale.

From an uninformed perspective, without knowledge of which activations are clean or poisoned, we can compute the overall variance (or standard error/deviation) of the summary statistic—in this case, the total persistence of 1-bars—across the entire sample. This is shown in the bottom right plot of Figure 10. Notably, we observe that this variance correlates with the absolute difference in total persistence between clean and poisoned activations. We analyzed their peaks and test if peaks in the overall variance are indicative of peaks in the absolute difference of total persistence using precision at $k$ (p@k) and testing for statistical significance using permutation tests. Results in Table 2 show that the two are strongly associated. Namely, peaks in total variance are a good indication of larger differences (peaks) in the absolute difference in several barcode summaries, and this effect is apparent among statistics for 1-bars rather than in 0-bars (see Table 5 for Mistral and LLaMA3_7B results).

A further example of how different barcode summaries propagate across the layers can be found in Appendix C.2.1 for

the Mistral model, showing the patterns for the mean deaths of 0-bars. Similar analyses for Phi3 and LLaMA3 8B models are found in Appendix C.2.2 and C.2.3, respectively.

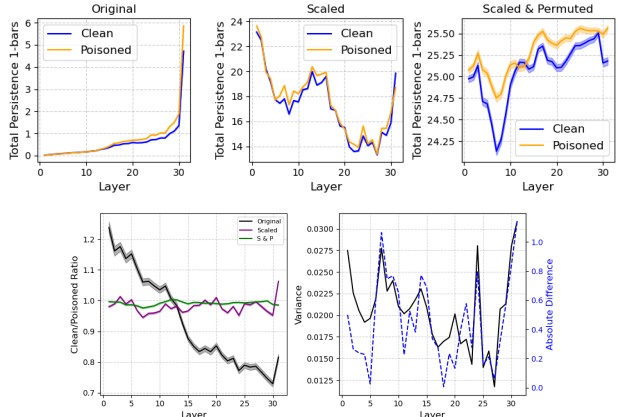

*Figure 10.* **Local analysis of consecutive layers for the total persistence of 1-bars. Top:** Comparisons of the average total persistence of 1-bars across 1000 samples for Mistral model for original (raw), scaled (normalized) and scaled & permuted activation data. **Bottom left:** Ratios of mean total persistence of 1-bars between clean and poisoned datasets for original, scaled and scaled & permuted activations. **Bottom right:** Overlaid plots of the overall variance of total persistence of 1-bars for clean and poisoned datasets combined and the absolute difference between mean total persistence of 1-bars for clean and poisoned datasets.

**Non-Consecutive Layers Analysis.** Expanding our analysis to activations collected from two non-consecutive layers, we apply the same barcode summary computations and scaling variations to the activations. Our assumption is that in neighboring layers, the model operates on similar groups of neurons, leading to element-wise interactions that construct meaningful topological features distinguishing clean from poisoned datasets. However, as we examine layer pairs that are farther apart, these distinctions in interactions between clean and poisoned activations become less pronounced.

Figure 11 illustrates this progression through the ratio of mean death times of 0-bars between clean and poisoned

*Table 2.* **Peak analysis.** Precision@$k$ for $k$=1, 3, and 5 largest peaks in total variance, and their precision in detecting the largest peaks in absolute difference between the two classes. *, ** correspond to $p$-values $<$.05 and .01, respectively.

|  | p@1 | p@3 | p@5 |
|---|---|---|---|
| Total persistence 0-bars | 0 | .33 | .4 |
| Total persistence 1-bars | 0 | .67* | .8** |
| Mean birth 1-bars | 1.0* | .33 | .8** |
| Mean death 1-bars | 1.0* | .33 | .8** |

activations as the layer interval increases. We observe that for layer intervals of 1 and 3, the ratios for scaled activations and scaled activations with permutation remain distinct from the control, indicating meaningful topological interactions. However, at an interval of 10 layers, the scaled and control settings show significant overlap, suggesting a diminishing effect of neighboring interactions. A similar pattern can be observed for other barcode summaries, such as the total persistence of 1-bars, which can be found in Appendix C.2.5.

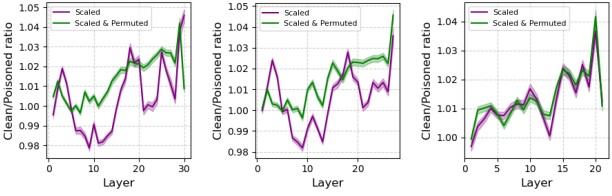

*Figure 11.* **Local analysis of non-consecutive layers for mean death of 0-bars.** Comparison of ratios between mean death of 0-bars for clean and poisoned datasets when considering topology pairs of layers at 1 (left), 3 (middle), and 10 (right) intervals apart.

## 5. Discussion

In this section, we discuss the broader implications of our work. We also raise important limitations of our study.

### 5.1. Intellectual Merit

Our study provides solid evidence that TDA approaches are sufficient to independently classify binary tasks using neural network activations and can achieve perfect accuracy without leveraging traditional activation-based methods. In contrast to previous approaches that aggregate entire activation vectors, we introduce a fine-grained micro-scale analysis, capturing topological structures where groups of neurons may form cycles (1-bars). These features are efficient and remain robust even under validation against strict control conditions. By carefully pruning redundant TDA-based features, we achieve high classification performance based on streamlined features with a more compact representation. Further refinement with SHAP analysis confirms that one or two key features are typically sufficient for essentially perfect classification.

Beyond classification, our results reveal how topological signatures of adversarial triggers become more pronounced in deeper layers. This insight moves beyond simple detection to revealing how adversarial manipulations may distort neural representations. Not only can we identify compromised activations, but we can also characterize the specific topological transformations that distinguish clean from poisoned states. Our results indicate that different adversarial conditions induce distinct deformations in topological structure, implying a broader applicability of our method across threat scenarios and in more general settings.

Our findings establish the following twofold potential for intellectual merit. First, by pinpointing how adversarial or biased inputs distort latent spaces, our approach lays the foundations for the construction of more robust and fair machine learning pipelines. Second, a micro-scale analysis captures structural details—where neuron interactions create higher order homology—that are often lost in large-scale aggregation, enriching the application of TDA in deep learning to provide richer interpretability than purely task-specific detectors.

Overall, our work advances both theoretical and applied perspectives in LLMs, NLP, and TDA. By revealing the persistent and interpretable topology and geometry of neuron-level interactions, it reinforces the position of topology as a powerful unifying framework for adversarial detection, representation learning, and interpretability in neural networks (Papamarkou et al., 2024).

### 5.2. Limitations

Our study and findings are restricted by the following limitations. A significant challenge associated with using PH is its computational expense. PH is computed with respect to a filtration, which makes the procedure inherently non-parallelizable and impossible to apply to very large datasets. We implemented random subsampling to compute proxy barcodes for the entire dataset, thus, our results are subject to sampling errors. However, subsampling has been well-studied in TDA; in particular, convergence results have been established (Chazal et al., 2014; Cao & Monod, 2022), so the sampling errors in our study are guaranteed to be bounded. Our analysis is also limited to two adversarial scenarios, leaving open the question of whether our observed topological signatures generalize to other forms of influence.

## 6. Conclusion

In this work, we applied PH to identify topological signatures of LLM representation spaces that are consistent across model families, sizes, and two adversarial conditions. Future work could investigate whether topological compression is a general property of misalignment and how it relates to model generalization (Stephenson et al., 2021); develop topology-aware robustness mechanisms (Brüel-Gabrielsson et al., 2020); or use persistent Morse theory (Bobrowski & Adler, 2014) and adapt cycle matching approaches (Reani & Bobrowski, 2022; García-Redondo et al., 2024) to further characterize representation spaces across behavioral modes.

## Impact Statement

This paper presents work whose goal is to advance the field of machine learning. There are many potential societal consequences of our work, none which we feel must be specifically highlighted here.

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

# A. Further Topological and Local Variance Interpretation

## A.1. Persistent Homology Barcode Statistics

To interpret the barcodes from Section 3.2, we extract key summary statistics that quantify the topological structure observed at each layer under both adversarial conditions.

From each 1-dimensional (1D) barcode, we gather intervals $(b_i, d_i)$ with $d_i > b_i > 0$ and define $\ell_i = d_i - b_i$. Forming a discrete distribution $p_i = \ell_i / \sum_j \ell_j$, the *persistent entropy* is

$$E = -\sum_i p_i \ln(p_i + \epsilon),$$

where $\epsilon$ is a small positive constant (e.g., $10^{-12}$) to ensure numerical stability. Higher $E$ indicates a more uniform distribution of lifetimes (no single interval dominates), whereas lower $E$ reflects a small number of long-lived intervals.

In addition to **entropy**, we compute the following summary statistics on dimension-1 bars:

- **Mean births (1-bars):** Average birth time $\bar{b}$

- **Mean deaths (1-bars):** Average death time $\bar{d}$

- **Mean persistence (1-bars):** Average lifetime $\overline{(d_i - b_i)}$

- **Number of 1-bars:** Count of finite intervals in dimension 1

We perform these computations for each barcode individually and then *average* over all barcodes in the same condition (*locked* or *elicited*) and (*clean* or *poisoned*).

A.1.1. EXTENDED PROMPT INJECTION (CLEAN VS. POISONED)

*Table 3.* **Dimension-1 persistent homology differences (clean − poisoned) in key metrics for three models across several layers.**
Positive values mean the *clean* condition has a higher value, while negative indicates *poisoned* is higher for that metric. All entries rounded to four decimals.

| Model | Layer | Mean births 1-bars_diff | Mean deaths 1-bars_diff | Mean persistence 1-bars_diff | Entropy 1-bars_diff | Number 1-bars_diff |
|---|---|---|---|---|---|---|
| LLaMA-3 (8B) | 1 | -0.0005 | -0.0006 | -0.0001 | 0.1665 | 86.9700 |
| | 8 | -0.0609 | -0.0608 | 0.0001 | 0.1213 | 79.5600 |
| | 16 | -0.3166 | -0.3249 | -0.0082 | 0.0188 | 17.9367 |
| | 24 | -0.9932 | -1.0256 | -0.0324 | 0.1595 | 80.0833 |
| | 32 | -18.3367 | -18.9290 | -0.5923 | 0.3348 | 192.4900 |
| Mistral (7B) | 1 | 0.0004 | 0.0004 | 0.0000 | 0.0172 | 3.7967 |
| | 8 | -0.0293 | -0.0295 | -0.0002 | 0.1485 | 118.9167 |
| | 16 | -0.2375 | -0.2421 | -0.0047 | 0.1938 | 154.7633 |
| | 24 | -0.5694 | -0.5815 | -0.0120 | 0.2070 | 153.9633 |
| | 32 | -14.7376 | -15.0558 | -0.3182 | 0.2239 | 166.4267 |
| Phi 3 (3.8B) | 1 | 0.0011 | 0.0009 | -0.0002 | 0.0101 | 4.3200 |
| | 8 | -0.4522 | -0.4675 | -0.0153 | 0.0888 | 59.0967 |
| | 16 | -1.7825 | -1.8293 | -0.0467 | 0.0696 | 48.7367 |
| | 24 | -6.2017 | -6.3448 | -0.1431 | 0.1665 | 116.9433 |
| | 32 | -1.0092 | -1.0503 | -0.0411 | -0.4857 | -136.1200 |

For **mean births** and **mean deaths**, all layers except layer 1 across models have negative differences, indicating that *poisoned* intervals emerge and die later in the filtration. The mean persistence is also higher for *poisoned* examples, especially in the middle layers. The entropy differences are also almost all positive (except Phi3 3.8's final layer), suggesting that *clean*

distributes its 1D lifetimes more evenly, while *poisoned* tends to concentrate total persistence in fewer cycles. Likewise, the number of 1-bars remains mostly positive, so *clean* produces more 1D holes overall.

Overall, *poisoned* conditions often yield fewer but more dominant dimension–1 features that appear later and persist longer, whereas *clean* spreads its total persistence across numerous shorter-lived cycles, resulting in higher entropy. Depending on the model, the specific layer where these features distinctions appear strongest differ but the nature of the difference is the same across models and layers.

A.1.2. SANDBAGGED MODEL (LOCKED VS. ELICITED)

*Table 4.* **Dimension-1 persistent homology differences (elicited − locked) for two models across multiple layers.** Positive values indicate that the *elicited* condition has higher values; negative means *locked* is higher for that metric.

| Model | Layer | Mean births 1-bars_diff | Mean deaths 1-bars_diff | Mean persistence 1-bars_diff | Entropy 1-bars_diff | Number 1-bars_diff |
|---|---|---|---|---|---|---|
| **LLaMA-3 (8B)** | 0 | -0.0127 | -0.0132 | -0.0005 | 0.0156 | 3.2400 |
| | 7 | -0.3425 | -0.3555 | -0.0130 | 0.0647 | 27.8600 |
| | 15 | -0.0476 | -0.0455 | 0.0021 | 0.2114 | 135.2900 |
| | 23 | -0.1168 | -0.1204 | -0.0037 | 0.0100 | 61.8766 |
| | 31 | -0.9750 | -1.0458 | -0.0707 | 0.0620 | 28.2800 |
| **Mistral (7B)** | 0 | -0.0053 | -0.0055 | -0.0002 | 0.0942 | 27.1533 |
| | 7 | -0.1925 | -0.1989 | -0.0064 | 0.0310 | 14.1066 |
| | 15 | 0.0393 | 0.0352 | -0.0041 | 0.0277 | 10.9300 |
| | 23 | 0.6722 | 0.7037 | 0.0315 | -0.0363 | -0.1900 |
| | 31 | 14.6450 | 15.2952 | 0.6503 | -0.0014 | 9.3233 |

For LLaMA3 8B , the **mean birth and death** differences are negative across all computed hidden layers (1, 8, 16, 24, 32). Note that layers are zero-indexed, meaning that layer 0 corresponds to the first hidden layer, layer 1. This indicates that, in the *locked* condition, 1D cycles exhibit larger (i.e., later) birth and death times compared to *elicited*. In other words, when locked, the 1D features tend to emerge "further out" in the filtration. The **mean persistence** difference between conditions is also negative (except layer 16), suggesting that locked cycles generally persist slightly longer on average. Entropy differences are positive, indicating that *elicited* exhibits a greater diversity or spread among the lifetimes of its 1D features. The **number of 1-bars** is positive (sometimes strongly so), meaning there are substantially more 1D features in the *elicited* condition.

We see similar results for Mistral 7B  with negative differences in births and deaths in earlier layers, implying that locked has larger birth/death times at those lower layers. However, the sign flips, with *elicited* displaying larger values for births, deaths, and persistence. Specifically, layer 32 shows a notably large positive difference (e.g., +14.64 for births, +15.29 for deaths), indicating that the final layer in *elicited* captures significantly later 1D cycles relative to locked. The **number of 1-bars** also tends to be higher in elicited at most layers, except for a minor negative at layer 23, again suggesting that elicited reveals a greater number of dimension–1 features.

### A.2. Local Dispersion Ratio Analysis

We analyze how local geometry in hidden-layer representation space differs between *clean* and multiple *poisoned* modes in six LLMs. We further classify poisoned prompts into three sub-types:

1. **Executed:** The injected request is recognized and carried out (indirect prompt injection).

2. **Refused:** The model identifies the injected content as malicious and issues a refusal, effectively "shutting down" any detailed elaboration.

3. **Ignored:** The model neither executes nor refuses, but effectively overlooks the injected prompt, proceeding as if it were absent.

For each final token's activation difference vector $\Delta\mathrm{Act}_\ell(x_i) \in \mathbb{R}^D$, we identify its $k$ nearest neighbors in layer $\ell$ and perform PCA on those points. Let $\lambda_1 \geq \cdots \geq \lambda_{D'}$ be the resulting eigenvalues. We define the *dispersion ratio* of $\Delta\mathrm{Act}_\ell(x_i)$ as

$$\frac{\sum_{j=2}^{D'} \lambda_j}{\lambda_1 + \epsilon},$$

where $\epsilon$ prevents division by zero. A higher ratio indicates that variance is more evenly spread among secondary directions, whereas a lower ratio implies most variance lies in a single dominant direction.

**Ablation: Clean vs. Clean, Poisoned vs. Poisoned, and Mixed.** To confirm that dispersion discrepancies primarily reflect true *clean* vs. *poisoned* distinctions rather than random partitioning or mixture effects, we performed three auxiliary comparisons:

1. **Clean vs. Clean**: Split the clean set into two subsets, ensuring no significant difference arises from sampling within the same class.

2. **Poisoned vs. Poisoned**: Applied the same procedure to poisoned data to assess within-class variability.

3. **Mixed vs. Mixed**: Randomly partitioned a combined pool of clean and poisoned samples into two balanced groups.

**Note on Statistical Methods:** For every layer in each subplot, we computed the dispersion ratio for both *clean* and the specified *poisoned* (or *refused*, *executed*, *ignored*) samples. We then conducted a Welch's $t$-test on these two groups (clean vs. poisoned/other), applying false-discovery rate (FDR) correction across layers. We also verified approximate normality via kernel density estimates (KDEs) for each groups. Plot markers with stars indicate layers where $p_{\mathrm{FDR}} < 0.05$, confirming a statistically significant difference in dispersion ratio. To select $k = 30$, we tested candidate neighborhood sizes across layers and models, measuring which $k$ produced the largest absolute difference in mean local dispersion ratio between clean and poisoned conditions.

A.2.1. DISCUSSION OF RESULTS

Figures 12 and 13 highlight that:

- **Early Layers (Layer 1–8):** Across all poisoning modes, the *clean* condition consistently shows a higher dispersion ratio, suggesting that the model initially allocates broader representational capacity for normal inputs.

- **Mid Layers (Layer 16):** This pattern often *flips*, with poisoned prompts (especially *executed* or *ignored*) exceeding the clean baseline, indicating the network is dedicating extra directions to elaborate or "embrace" these injected requests. Conversely, *refused* prompts typically exhibit reduced dispersion, mapping disallowed content into a lower-variance region.

Interestingly, our findings align with the results in Stephenson et al. (2021), which indicate that memorization tends to emerge in deeper layers where the effective dimensionality shrinks. Consistent with that view, we observe that *executed* or *ignored* prompts show a higher dispersion in mid-layers, implying the model invests additional capacity there for those injected instructions. Meanwhile, a *refused* request is routed into a more compressed region, effectively "shutting down" further representational expansion. In this sense, deeper layers may provide a setting where the network can more sharply discriminate or overfit certain inputs—supporting the idea that final layers reflect a gradually compressed, yet strategically focused representation space.

**A.3. Cosine Distance of Representations**

We analyze the difference representations $\Delta\mathrm{Act}_\ell(x_i) \in \mathbb{R}^D$ for corresponding pairs of *clean* and *poisoned* inputs. Specifically, for each model and layer, we load up to five pairs of *clean* and *poisoned* activation files, compute the difference between the activations for each pair, and concatenate these differences. From these differences, we draw equal-size subsamples of 5,000 vectors. For each layer and comparison condition, we compute the mean pairwise cosine distance

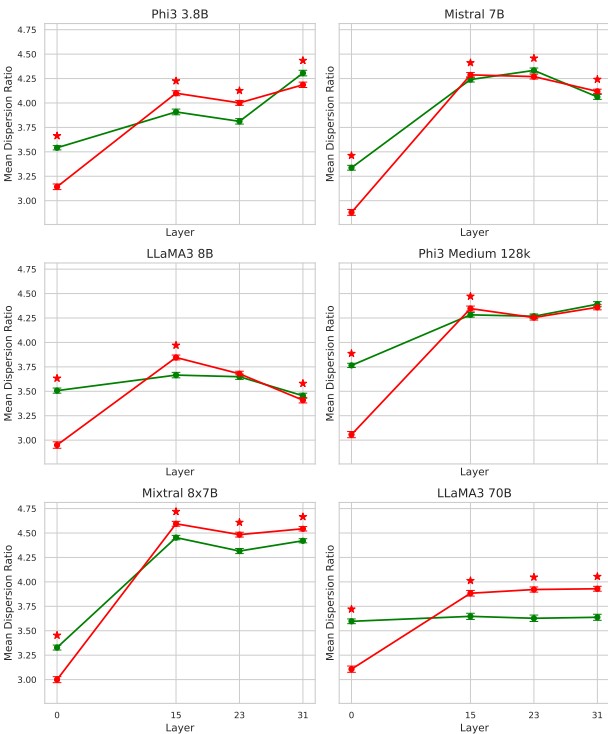

*Figure 12.* **Layer-wise Dispersion Ratio for Clean vs. Poisoned Examples.** The green and red lines depict mean dispersion ratios for *clean* and *poisoned* inputs, respectively, at different layer depths. Error bars around each point represent ±1 standard error of the mean (SEM). In early layers (left side), *clean* data consistently has higher dispersion on average, whereas in mid-layers (center), *poisoned* surpasses the clean baseline, indicating a re-distribution of representational capacity for the injected prompts. Layers where the difference is statistically significant ($p_{\mathrm{FDR}} < 0.05$) are marked with a red asterisk above the higher mean value.

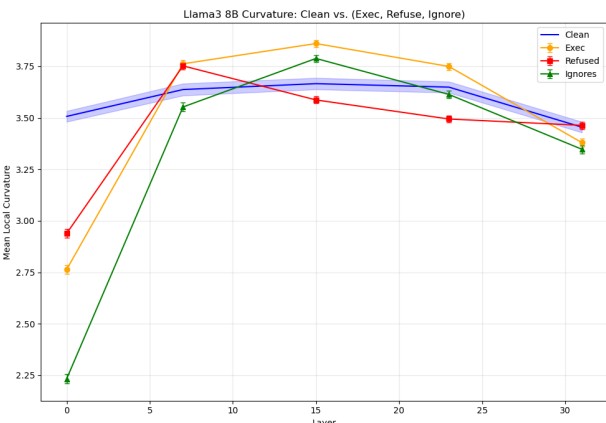

*Figure 13.* ***LLaMA3_7B* Dispersion Ratio: Clean vs. Executed, Refused, and Ignored Prompts.** The horizontal axis indicates layer depth, while the vertical axis represents the mean dispersion ratio. The blue curve (with confidence band) corresponds to *clean* inputs; orange, red, and green curves denote *executed*, *refused*, and *ignored* poisoned prompts, respectively. Notably, *refused* prompts show an early jump but then collapse below the clean baseline, whereas *executed* and *ignored* surpass it around mid-layers, highlighting distinct representational regimes.

within each subsample. Because cosine distance is scale-invariant, we do not normalize these difference representations. We perform four comparison conditions: *clean* vs. *poisoned*, *clean* vs. *clean* (where *clean* samples are split in half), *poisoned* vs. *poisoned* (where *poisoned* samples are split in half), and *mixed* vs. *mixed* (where two separate *mixed* subsamples are

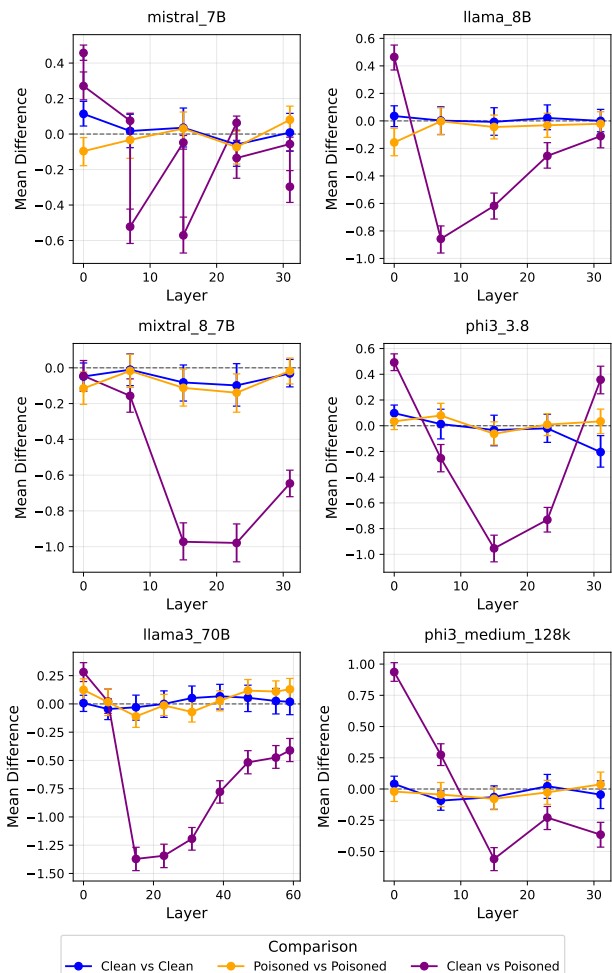

*Figure 14.* **Ablation of Dispersion Ratio Differences (Clean vs. Clean, Poisoned vs. Poisoned, Mixed vs. Mixed).** Each plot shows the *difference* in mean dispersion ratio (clean minus poisoned). Positive values indicate that the clean subset exhibits higher dispersion, whereas negative values reflect a more dispersed poisoned subset.

created, each containing half *clean* and half *poisoned* differences). For each comparison, we generate two distributions of mean pairwise intra-*class* distances (or inter-*class* in the *clean* vs *poisoned* case) using 3 bootstrap iterations. We then apply Welch's $t$-test to these distributions to assess whether they diverge significantly.

Empirically, *poisoned* difference representations typically exhibit a higher mean cosine distance in deeper layers, indicating a more "spread-out" or heterogeneous arrangement of their difference vectors, much as we observed in the curvature analysis. *Clean* data, by contrast, remains comparatively tightly clustered, implying less dispersion in its difference space. Interestingly, *LLaMA3_70B* displays similar characteristics in the early and final layers but poisoned representations have a noticeable smaller cosine distance in middle layers. This may reflect the ability of larger architectures to better partition representation space across the network before re-expanding in later layers.

## B. Further Details of Global Layer-Wise Analysis

### B.1. Pipeline

We describe in more detail the pipeline in Figure 3 in the main text. Recall that our aim here was showcasing that topological signatures effectively capture distinctions between representations under normal or adversarial conditions, and to provide an interpretation of the reason behind such difference in terms of the "shape" of the latent representations.

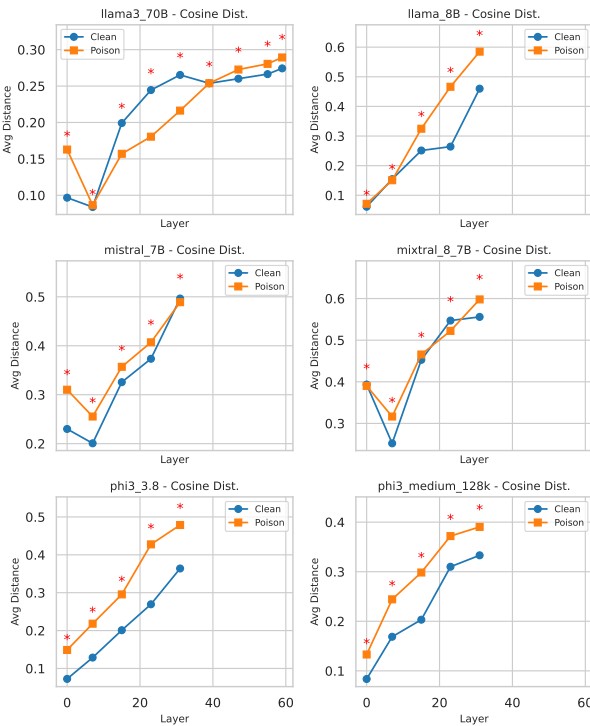

*Figure 15.* **Cosine Distance of Difference Representations Across Layers.** Each panel shows mean within-class distances (clean vs. poisoned) for the difference representations (*poisoned/clean pass* minus *baseline*), where higher values reflect greater variation among samples. Stars denote layers with significant differences.

We use RIPSER (Bauer, 2021) to compute barcodes, which is based on Vietoris–Rips filtrations (see Section 2.1). The computational constraints of PH make it impossible to compute the barcode of any of our two datasets (clean vs. poisoned or locked vs. elicited). Therefore, we leverage subsampling approaches (e.g., Chazal et al. (2015)) and compute barcodes from $K = 600$ subsamples $\{x_{i_1,\ell}, \ldots, x_{i_k,\ell}\} \subset \mathbb{R}^D$ with size $k = 1000$, of the representations per layer $1 \leq \ell \leq L$. From these, 300 are taken from normal activations and 300 from adversarial activations. We use these as proxies for the topology of the whole space.

Following Ali et al. (2023), we vectorize these barcodes as 41-dimensional feature vectors, which we call *barcode summaries*. These include 35 statistics derived from a $7 \times 5$ grid of {mean, minimum, first quartile, median, third quartile, maximum, standard deviation} × {death of 0-bars, birth of 1-bars, death of 1-bars, persistence of 1-bars, ratio birth/death of 1-bars}; as well as the total persistence (i.e., sum of the lengths of all bars in the barcode), number of bars, and persistent entropy (Chintakunta et al., 2015; Rucco et al., 2016) defined in Appendix A.1 for 0- and 1-bars. We reduce the dimensionality case-by-case, by eliminating highly correlated features (above a threshold of 0.5) through cross-correlation analysis.

For exploratory analysis, we apply PCA and compute CCA loadings to measure feature correlations with the principal components. A logistic regression model is then used for classification, and Shapley values (Lipovetsky & Conklin, 2001) are computed to evaluate feature importance. Shapley values, derived from cooperative game theory, quantify the contribution of each feature to model predictions by measuring its influence in shifting predictions from a baseline (e.g., 0.5 for logistic regression), providing an interpretable, feature-level analysis of predictive impact.

## B.2. Results: Clean vs. poisoned

### B.2.1. MISTRAL WITH EUCLIDEAN DISTANCE

We include more comprehensive results including layers 1, 8, 16, 23 and 32 for the running example of Section 4.1.

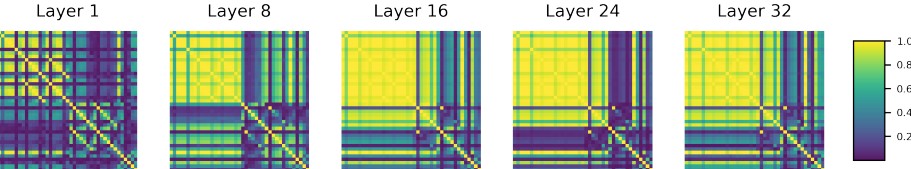

*Figure 16.* **Mistral with Euclidean distance: Cross-correlation matrices for the barcode summaries for clean vs. poisoned activations.** Growing block of correlated features appears in the cross-correlation matrix of the barcode summaries for layers 1, 8, 16, 24, and 32.

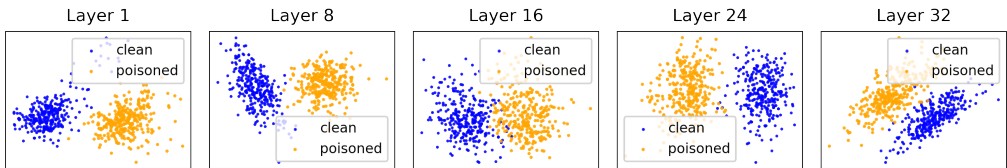

*Figure 17.* **Mistral with Euclidean distance: PCA of barcode summaries of clean vs. poisoned activations**. Clear distinction appears in the projection onto the two first principal components from the PCA of the pruned barcode summaries for layers 1, 8, 16, 24, and 32. The explained variance of is 0.39, 0.51, 0.53, 0.60 and 0.49, respectively.

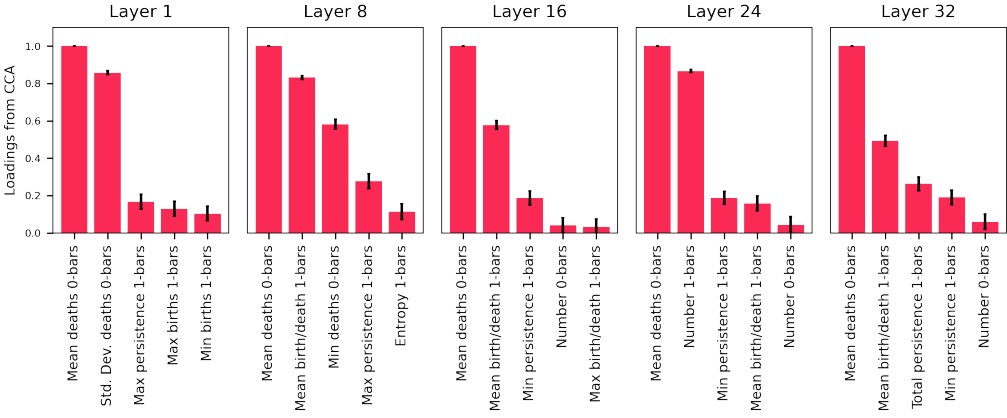

*Figure 18.* **Mistral with Euclidean distance: CCA loadings for clean vs. poisoned activations (Mistral with cosine distance)**. Loadings of the 5 most important contributions to the first canonical variable of the CCA on the pruned barcode summaries show that the mean of the death of 0-bars is significantly correlated with the first two principal components of the PCA across all layers.

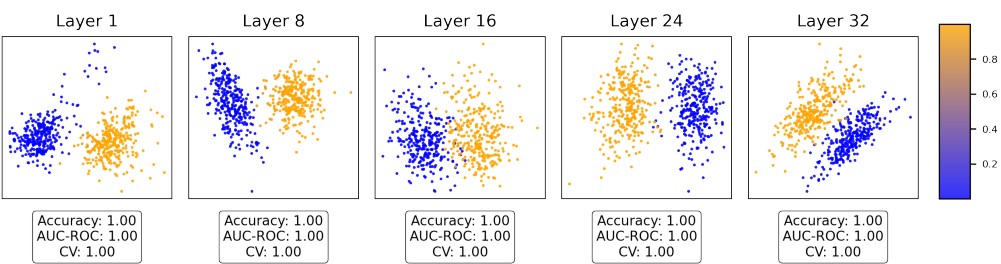

*Figure 19.* **Mistral with Euclidean distance: Logistic regression for clean vs. poisoned activations.** Prediction of a logistic regression trained on a 70/30 train/test split of the pruned barcode summaries, plotted on the projection onto the two first principal components for visualization purposes. Accuracy and AUC–ROC tested on the test data, and 5-fold cross validation on train data are presented for each model, showcasing the outstanding performance of all models.

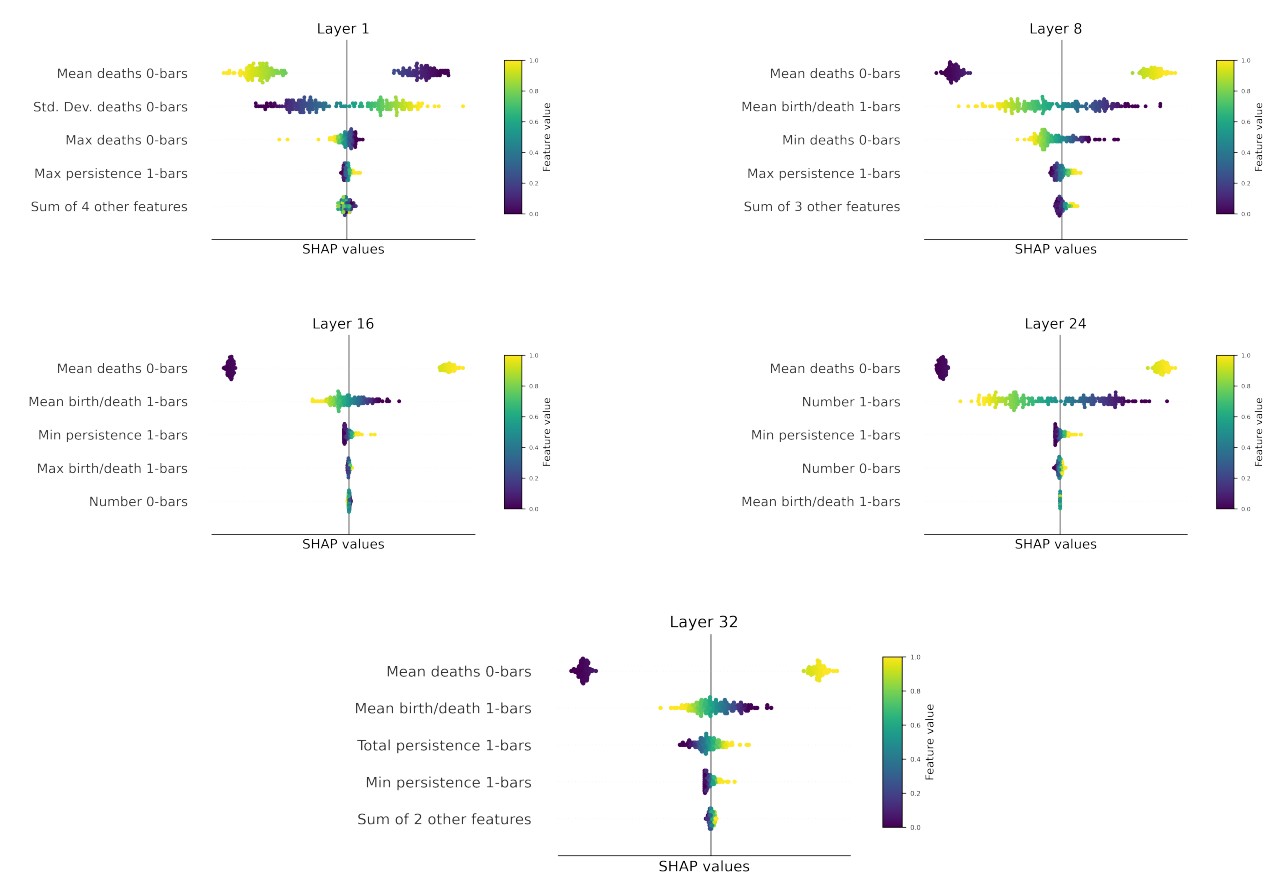

*Figure 20.* **Mistral with Euclidean distance: SHAP analysis for clean vs. poisoned activations.** Beeswarm plot of the SHAP values for the logistic regression trained on the pruned barcode summaries for layer 1, 8, 16, 24, and 32. The mean of the deaths of 0-bars appears as the most impactful feature in the prediction of the model, shifting predictions to "clean" when the value of the feature is lower for layers 8, 16, 23 and 32, and to "poisoned" when it is higher. The opposite phenomenon is observed in layer 0.

### B.2.2. MISTRAL WITH COSINE DISTANCE

We present the results of the analysis outlined in Figure 3 and Appendix B.1 for Mistral, but computing the barcodes of the Vietoris–Rips filtration using the cosine distance between activations. We prefer to include the result for the Euclidean distance in the main text as they provide a clearer interpretation. Main differences with the Euclidean case are that the presence of high correlated features seems to increase, particularly for middle layer of the model; and that there is no flip in the sign of the correlation between the value of the mean of deaths of 0-bars in the beeswarm plot of the SHAP values, meaning that lower values of this feature consistently push the prediction for an input barcode to be "clean."

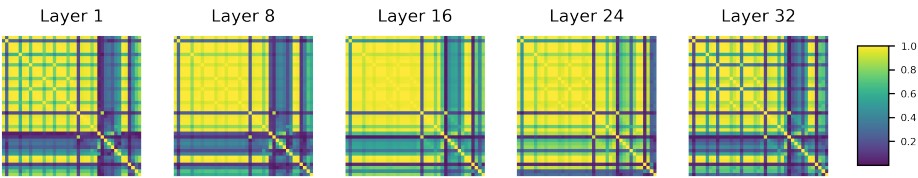

*Figure 21.* **Mistral with cosine distance: Cross-correlation matrices for the barcode summaries for clean vs. poisoned activations.** Growing block of correlated features appears in the cross-correlation matrix of the barcode summaries for layers 1, 8, 16, 24, and 32. Correlation is higher and more prominent from earlier layers, opposed to the analysis with Euclidean distance.

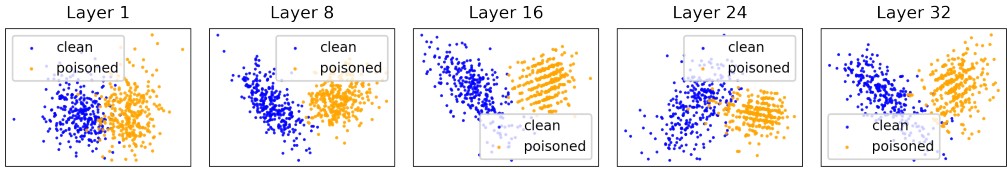

*Figure 22.* **Mistral with cosine distance: PCA of barcode summaries of clean vs. poisoned activations**. Clear distinction appears in the projection onto the two first principal components from the PCA of the pruned barcode summaries for layers 1, 8, 16, 24, and 32.

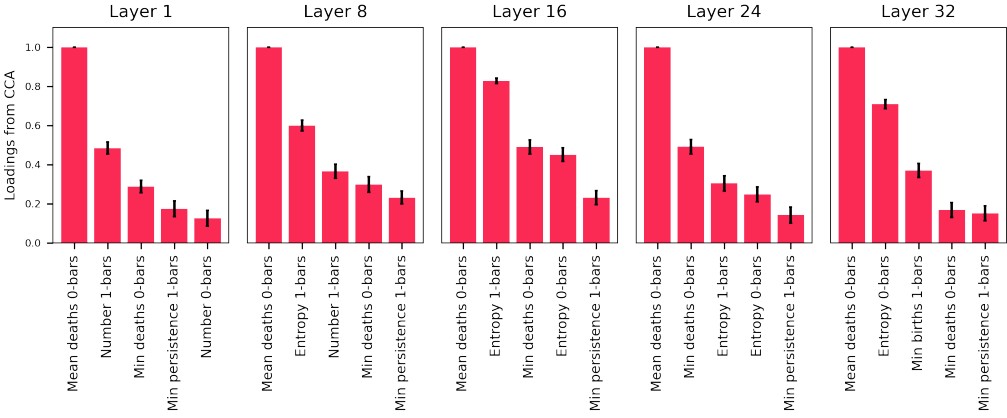

*Figure 23.* **Mistral with cosine distance: CCA loadings for clean vs. poisoned activations**. Loadings of the 5 most important contributions to the first canonical variable of the CCA on the pruned barcode summaries show that the mean of the death of 0-bars is significantly correlated with the first two principal components of the PCA across all layers.

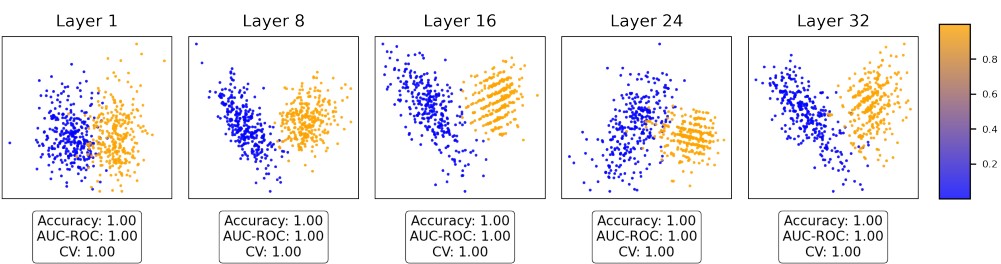

*Figure 24.* **Mistral with cosine distance: Logistic regression for clean vs. poisoned activations.** Prediction of a logistic regression trained on a 70/30 train/test split of the pruned barcode summaries, plotted on the projection onto the two first principal components for visualization purposes. Accuracy and AUC–ROC tested on the test data, and 5-fold cross validation on train data are presented for each model, showcasing the outstanding performance of all models.

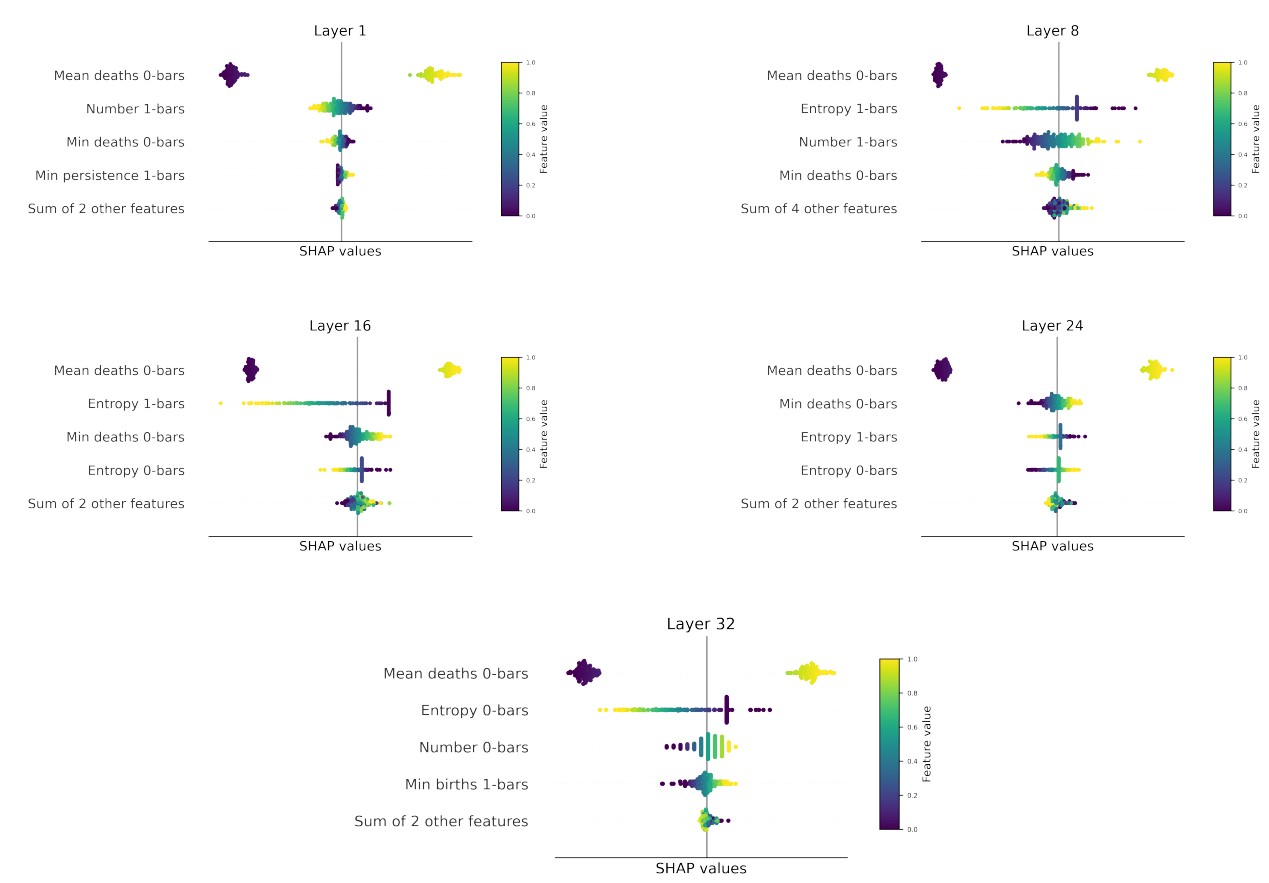

*Figure 25.* **Mistral with cosine distance: SHAP analysis for clean vs. poisoned activations.** Beeswarm plot of the SHAP values for the logistic regression trained on the pruned barcode summaries for layer 1, 8, 16, 24, and 32. The mean of the deaths of 0-bars appears as the most impactful feature in the prediction of the model, shifting predictions to "clean" when the value of the feature is lower for all layers, opposed to the models with Euclidean distance.

### B.2.3. LLAMA 3 WITH EUCLIDEAN DISTANCE

We provide the results of the analysis depicted in Figure 3 including layers 1, 8, 16, 23 and 32 for the Llama 3 where barcodes are computed using the Euclidean distance in the representation space. We observe very similar results to the ones obtained with Mistral, indicating a consinstency across models of the topological deformations of adversarial influence via XPIA (see Section 3.1).

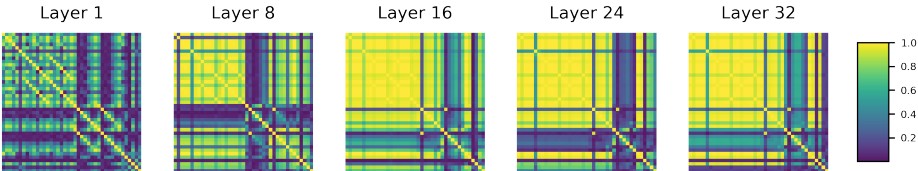

*Figure 26.* **Llama with Euclidean distance: Cross-correlation matrices for the barcode summaries for clean vs. poisoned activations.** Growing block of correlated features appears in the cross-correlation matrix of the barcode summaries for layers 1, 8, 16, 24, and 32. Correlations in layer 1 are lower than with Mistral, see Figure 16.

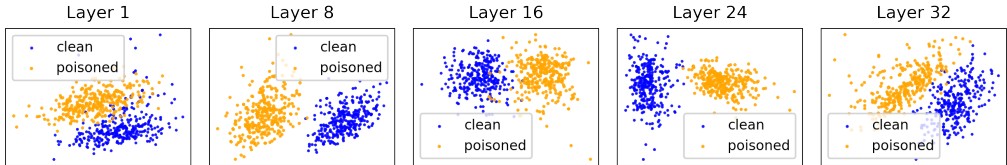

*Figure 27.* **Llama with Euclidean distance: PCA of barcode summaries of clean vs. poisoned activations**. Clear distinction appears in the projection onto the two first principal components from the PCA of the pruned barcode summaries for layers 1, 8, 16, 24, and 32. The explained variance of is 0.39, 0.51, 0.53, 0.60 and 0.49, respectively.

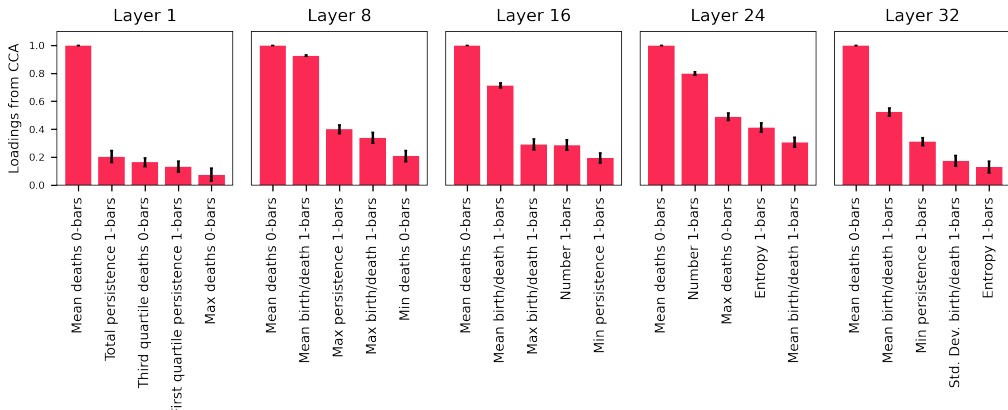

*Figure 28.* **Llama with Euclidean distance: CCA loadings for clean vs. poisoned activations**. Loadings of the 5 most important contributions to the first canonical variable of the CCA on the pruned barcode summaries show that the mean of the death of 0-bars is significantly correlated with the first two principal components of the PCA across all layers.

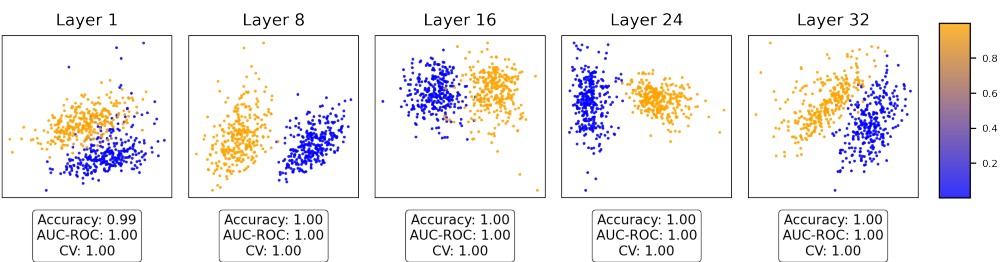

*Figure 29.* **Llama with Euclidean distance: Logistic regression for clean vs. poisoned activations.** Prediction of a logistic regression trained on a 70/30 train/test split of the pruned barcode summaries, plotted on the projection onto the two first principal components for visualization purposes. Accuracy and AUC–ROC tested on the test data, and 5-fold cross validation on train data are presented for each model, showcasing the outstanding performance of all models.

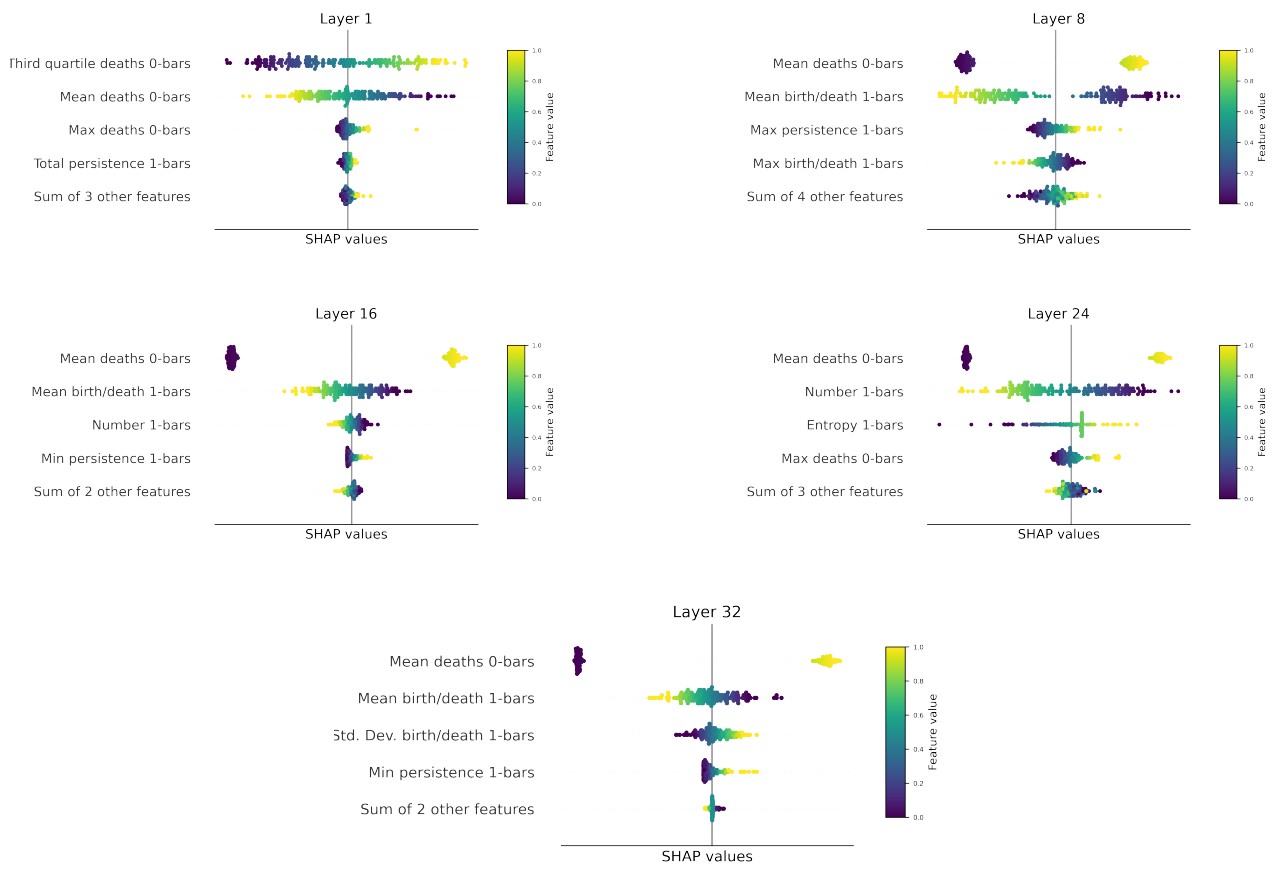

*Figure 30.* **Llama with Euclidean distance: SHAP analysis for clean vs. poisoned activations.** Beeswarm plot of the SHAP values for the logistic regression trained on the pruned barcode summaries for layer 1, 8, 16, 24, and 32. The mean of the deaths of 0-bars appears as the most impactful feature in the prediction of the model, shifting predictions to "clean" when the value of the feature is lower for all layers, contrary to the results with Euclidean distance.

### B.2.4. LLAMA 3 WITH COSINE DISTANCE

We provide the results of the analysis depicted in Figure 3 including layers 1, 8, 16, 23 and 32 for the Llama 3 where barcodes are computed using the cosine distance in the representation space.

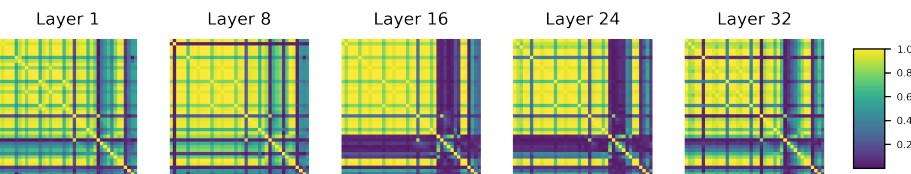

*Figure 31.* **Llama with cosine distance: Cross-correlation matrices for the barcode summaries for clean vs. poisoned activations.** Growing block of correlated features appears in the cross-correlation matrix of the barcode summaries for layers 1, 8, 16, 24, and 32.

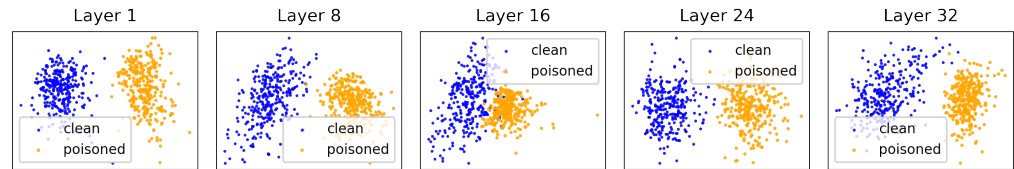

*Figure 32.* **Llama with cosine distance: PCA of barcode summaries of clean vs. poisoned activations**. Clear distinction appears in the projection onto the two first principal components from the PCA of the pruned barcode summaries for layers 1, 8, 16, 24 and 32.

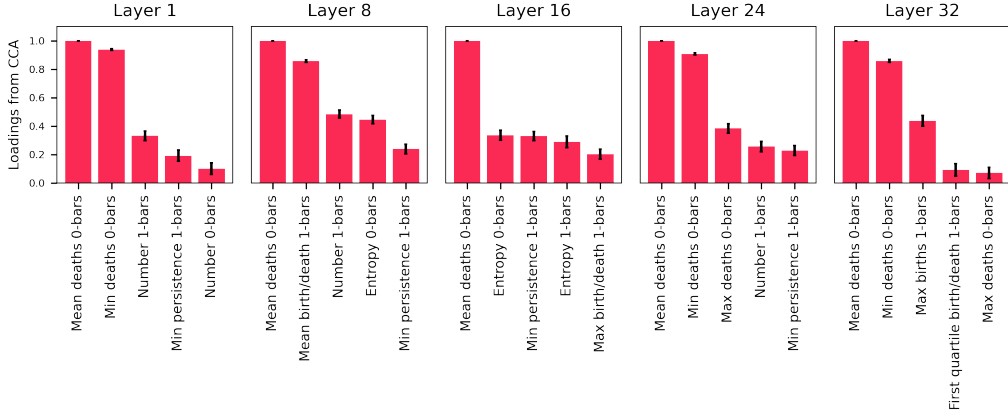

*Figure 33.* **Llama with cosine distance: CCA loadings for clean vs. poisoned activations**. Loadings of the 5 most important contributions to the first canonical variable of the CCA on the pruned barcode summaries show that the mean of the death of 0-bars is significantly correlated with the first two principal components of the PCA across all layers.

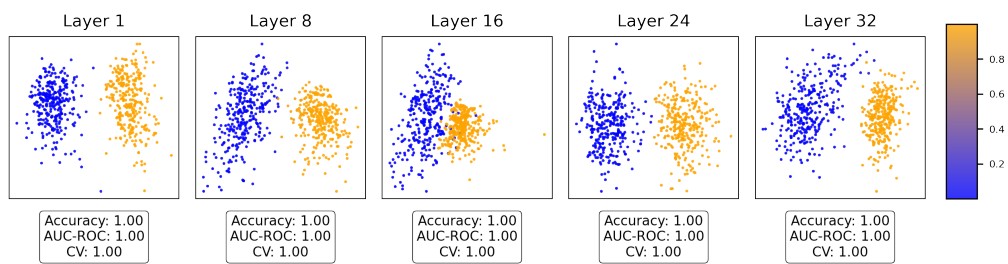

*Figure 34.* **Llama with cosine distance: Logistic regression for clean vs. poisoned activations.** Prediction of a logistic regression trained on a 70/30 train/test split of the pruned barcode summaries, plotted on the projection onto the two first principal components for visualization purposes. Accuracy and AUC–ROC tested on the test data, and 5-fold cross validation on train data are presented for each model, showcasing the outstanding performance of all models.

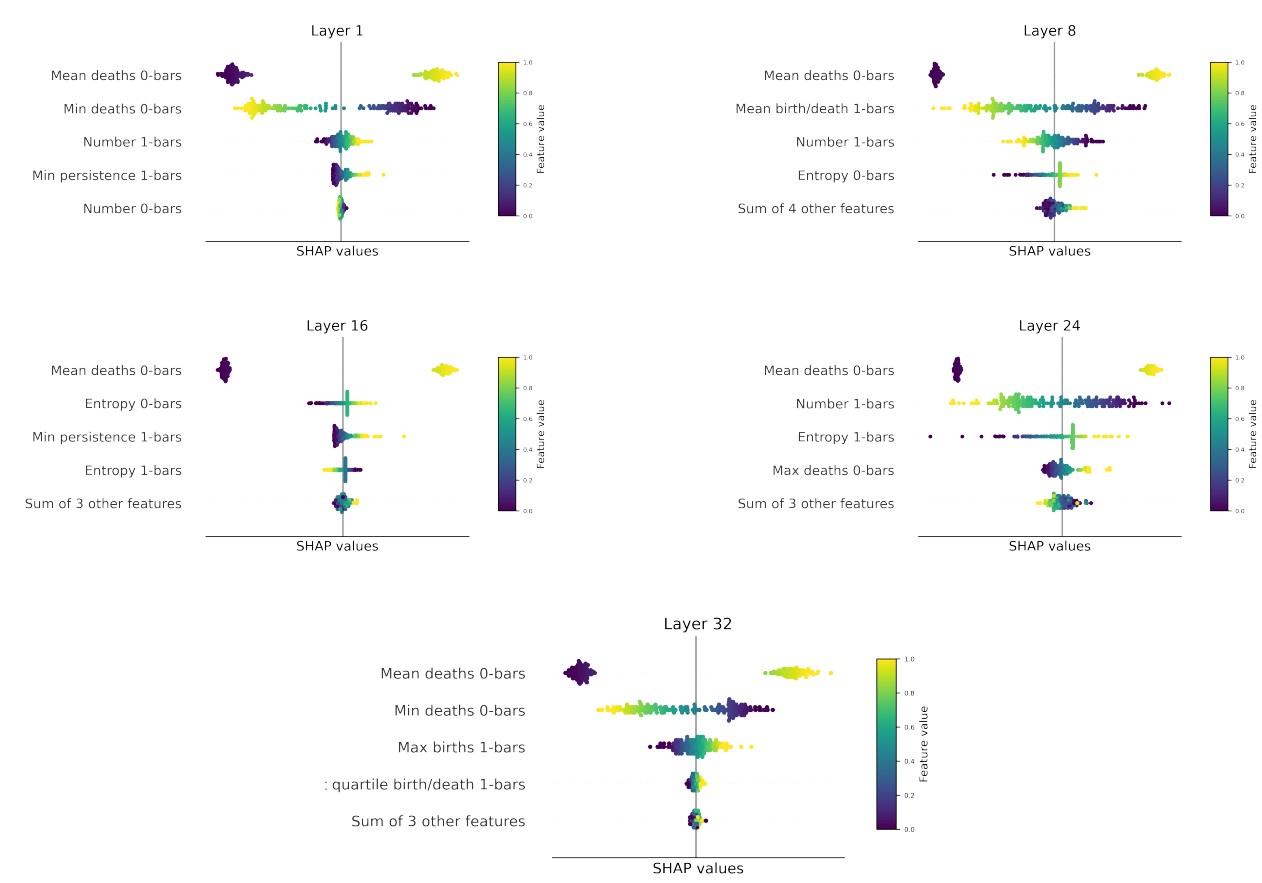

*Figure 35.* **Llama with cosine distance: SHAP analysis for clean vs. poisoned activations.** Beeswarm plot of the SHAP values for the logistic regression trained on the pruned barcode summaries for layer 1, 8, 16, 24, and 32. The mean of the deaths of 0-bars appears as the most impactful feature in the prediction of the model, shifting predictions to "clean" when the value of the feature is lower for layers 16 and 32, and to "poisoned" when it is higher. The opposite phenomenon is observed in layer 0.

### B.2.5. PHI 3 WITH EUCLIDEAN DISTANCE

We provide the results of the analysis depicted in Figure 3 including layers 1, 8, 16, 23, and 32 for Phi 3 where barcodes are computed using the Euclidean distance in the representation space.

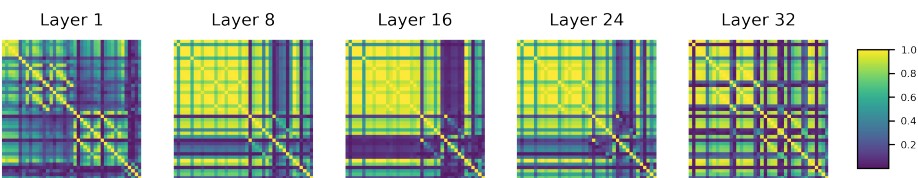

*Figure 36.* **Phi with Euclidean distance: Cross-correlation matrices for the barcode summaries for clean vs. poisoned activations.** Growing block of correlated features appears in the cross-correlation matrix of the barcode summaries appears in the middle layers (layers 1, 8, 16, 24, and 32 are shown).

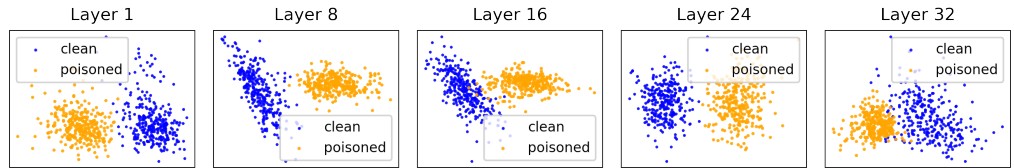

*Figure 37.* **Phi with Euclidean distance: PCA of barcode summaries of clean vs. poisoned activations**. Clear distinction appears in the projection onto the two first principal components from the PCA of the pruned barcode summaries for layers 1, 8, 16, 24, and 32.

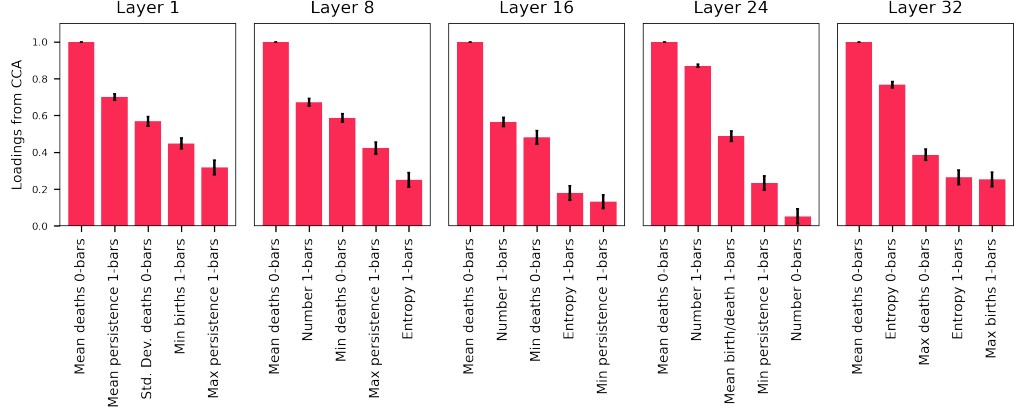

*Figure 38.* **Phi with Euclidean distance: CCA loadings for clean vs. poisoned activations**. Loadings of the 5 most important contributions to the first canonical variable of the CCA on the pruned barcode summaries show that the mean of the death of 0-bars is significantly correlated with the first two principal components of the PCA across all layers.

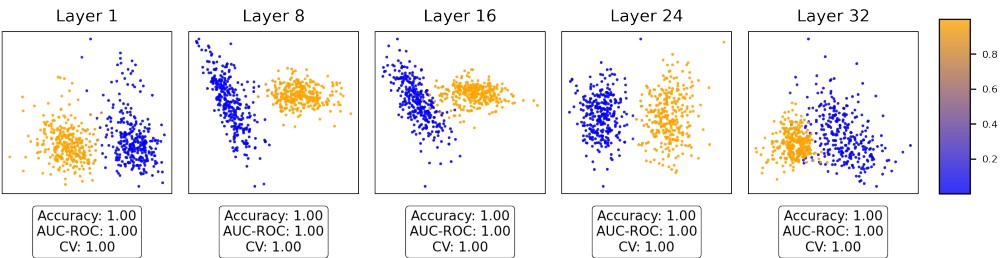

*Figure 39.* **Phi with Euclidean distance: Logistic regression for clean vs. poisoned activations.** Prediction of a logistic regression trained on a 70/30 train/test split of the pruned barcode summaries, plotted on the projection onto the two first principal components for visualization purposes. Accuracy and AUC–ROC tested on the test data, and 5-fold cross validation on train data are presented for each model, showcasing the outstanding performance of all models.

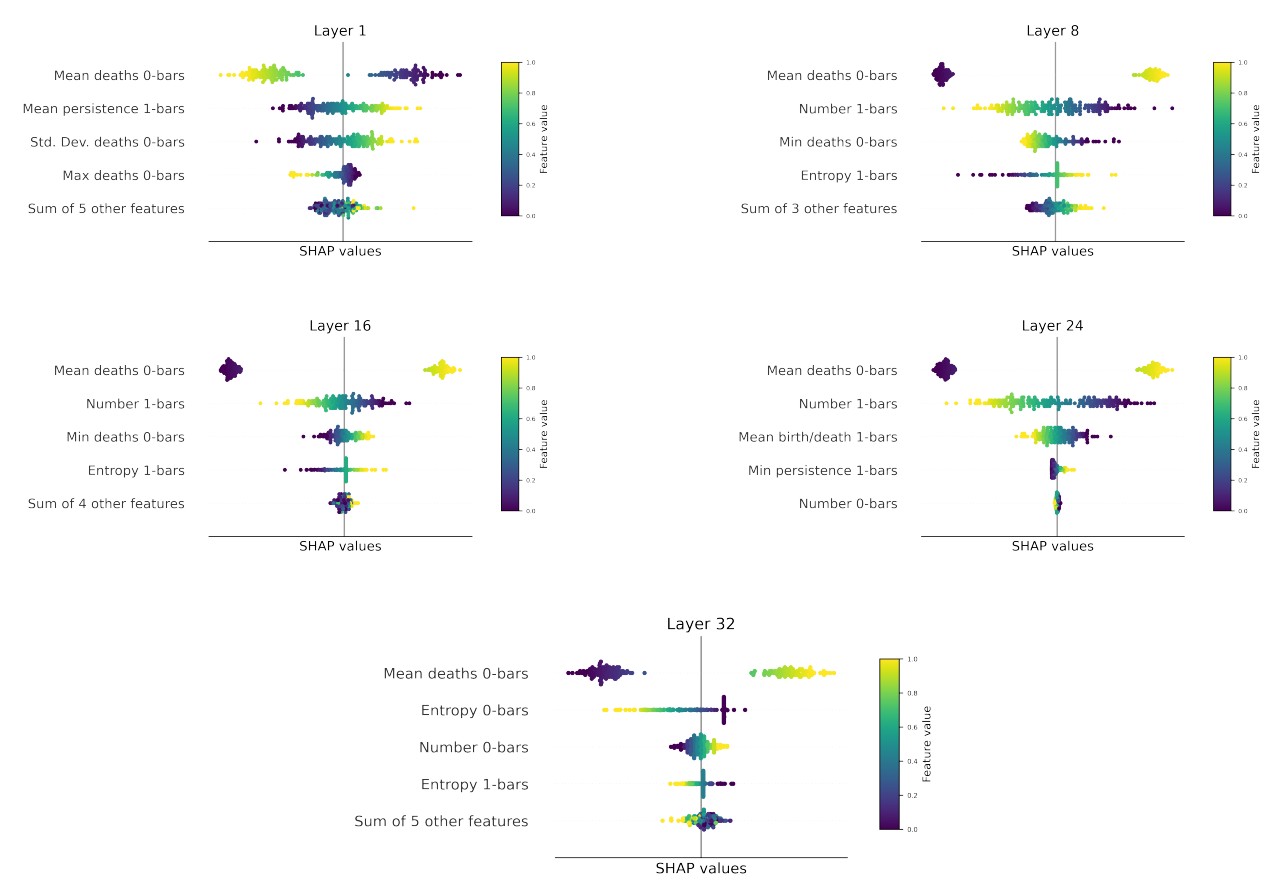

*Figure 40.* **Phi with Euclidean distance: SHAP analysis for clean vs. poisoned activations.** Beeswarm plot of the SHAP values for the logistic regression trained on the pruned barcode summaries for layer 1, 8, 16, 24, and 32. The mean of the deaths of 0-bars appears as the most impactful feature in the prediction of the model, shifting predictions to "clean" when the value of the feature is lower for all layers, contrary to the results with Euclidean distance.

### B.2.6. PHI 3 WITH COSINE DISTANCE

We provide the results of the analysis depicted in Figure 3 including layers 1, 8, 16, 23, and 32 for Phi 3 where barcodes are computed using the cosine distance in the representation space.

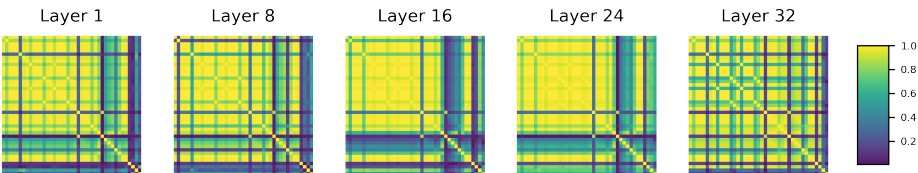

*Figure 41.* **Phi with cosine distance: Cross-correlation matrices for the barcode summaries for clean vs. poisoned activations.** Growing block of correlated features appears in the cross-correlation matrix of the barcode summaries appears in the middle layers (layers 1, 8, 16, 24, and 32 are shown). Higher correlations than in the Euclidean analysis appear particularly for early layers, some of those correlations are lost in latter layers.

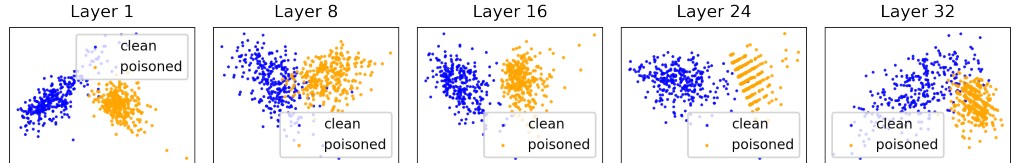

*Figure 42.* **Phi with cosine distance: PCA of barcode summaries of clean vs. poisoned activations**. Clear distinction appears in the projection onto the two first principal components from the PCA of the pruned barcode summaries for layers 1, 8, 16, 24, and 32.

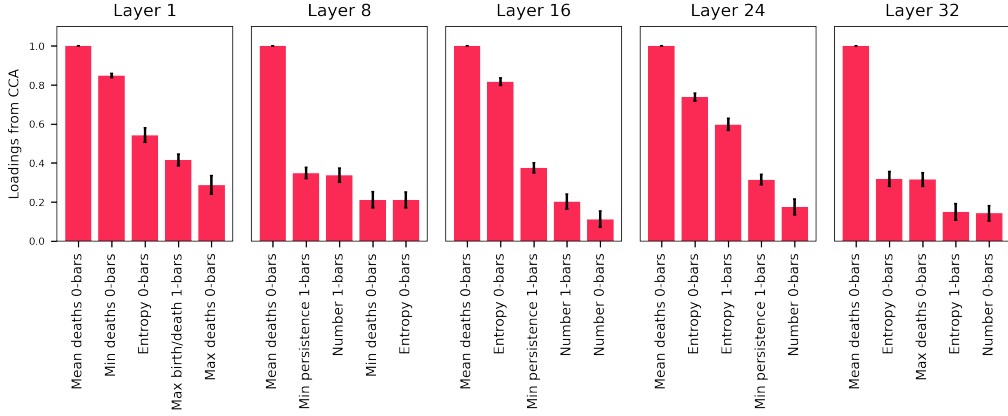

*Figure 43.* **Phi with cosine distance: CCA loadings for clean vs. poisoned activations**. Loadings of the 5 most important contributions to the first canonical variable of the CCA on the pruned barcode summaries show that the mean of the death of 0-bars is significantly correlated with the first two principal components of the PCA across all layers.

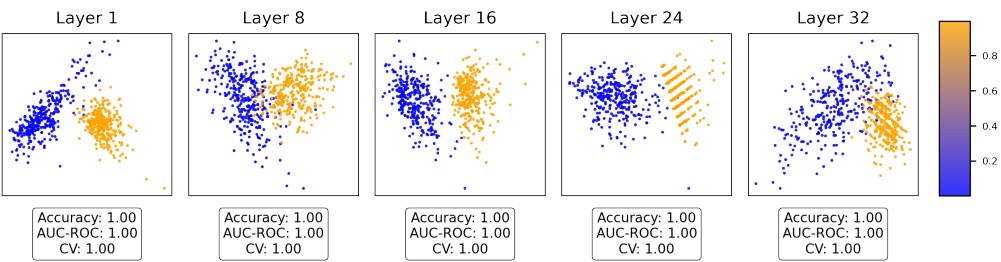

*Figure 44.* **Phi with cosine distance: Logistic regression for clean vs. poisoned activations.** Prediction of a logistic regression trained on a 70/30 train/test split of the pruned barcode summaries, plotted on the projection onto the two first principal components for visualization purposes. Accuracy and AUC–ROC tested on the test data, and 5-fold cross validation on train data are presented for each model, showcasing the outstanding performance of all models.

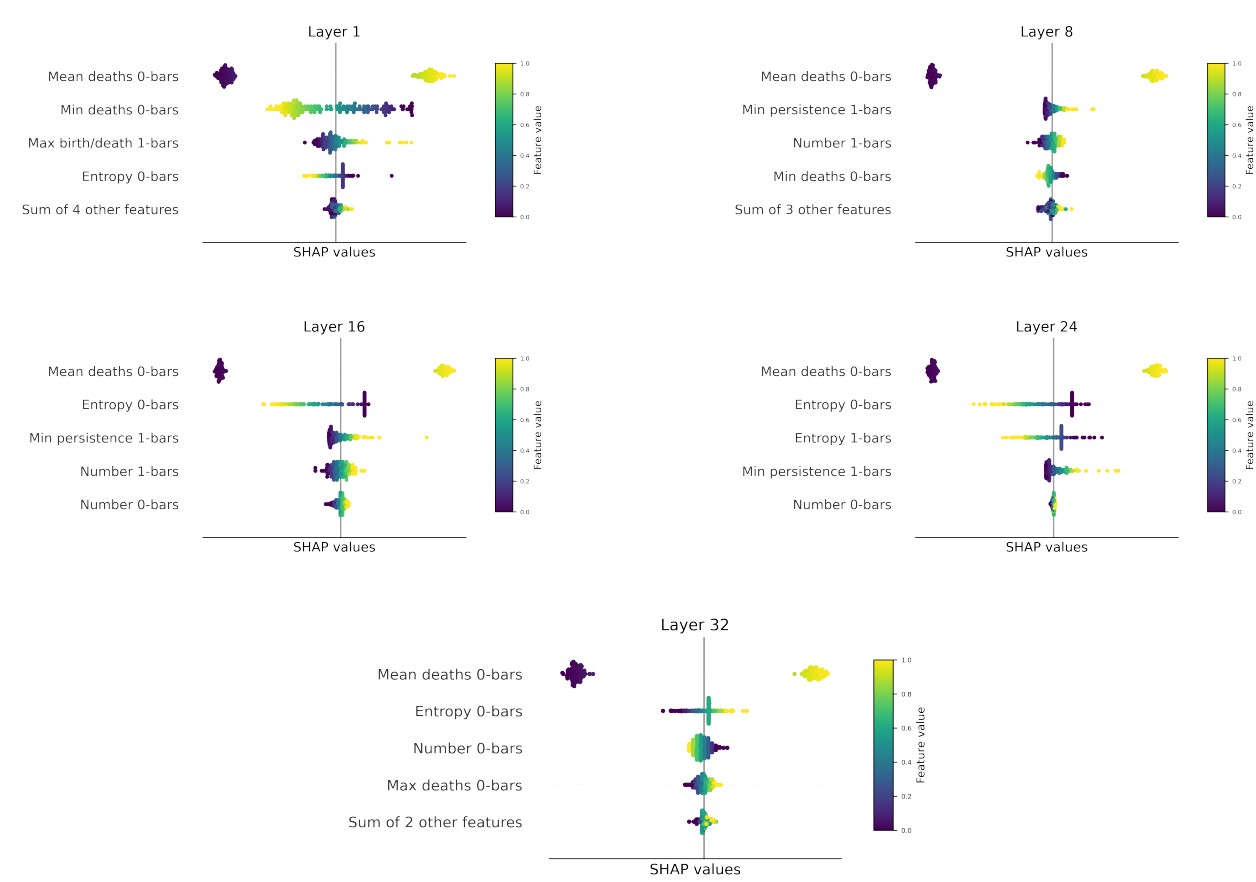

*Figure 45.* **Phi with cosine distance: SHAP analysis for clean vs. poisoned activations.** Beeswarm plot of the SHAP values for the logistic regression trained on the pruned barcode summaries for layer 1, 8, 16, 24, and 32. The mean of the deaths of 0-bars appears as the most impactful feature in the prediction of the model, shifting predictions to "clean" when the value of the feature is lower for all layers, contrary to the results with Euclidean distance.

## B.3. Results locked vs. elicited

### B.3.1. MISTRAL MODEL

We include the results of the global analysis in Figure 3 for the locked vs. elicited dataset. There are two main differences with previous results: the block of high correlated features presents a less clear trend and is more faint in layer 16, resulting

in the need of more features in the analysis; and the mean death of the 0-bars changes the sign of its influence in classifying locked and elicited models across layers. However the distinction in the PCA of the barcode summaries remains clear and the logistic regression still achieves perfect performance, despite the analysis resulting a bit less straightforward.

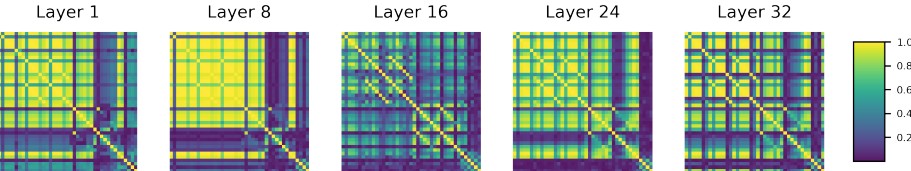

*Figure 46.* **Mistral with Euclidean distance: Cross-correlation matrices for the barcode summaries for locked vs. elicited activations.** Growing block of correlated features appears in the cross-correlation matrix of the barcode summaries for layers 1, 8, 16, 24, and 32.

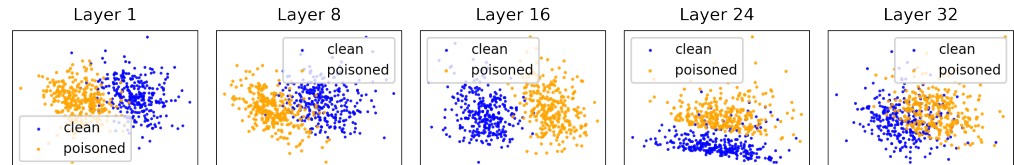

*Figure 47.* **Mistral with Euclidean distance: PCA of barcode summaries of locked vs. elicited activations.** Clear distinction appears in the projection onto the two first principal components from the PCA of the pruned barcode summaries for layers 1, 8, 16, 24, and 32.

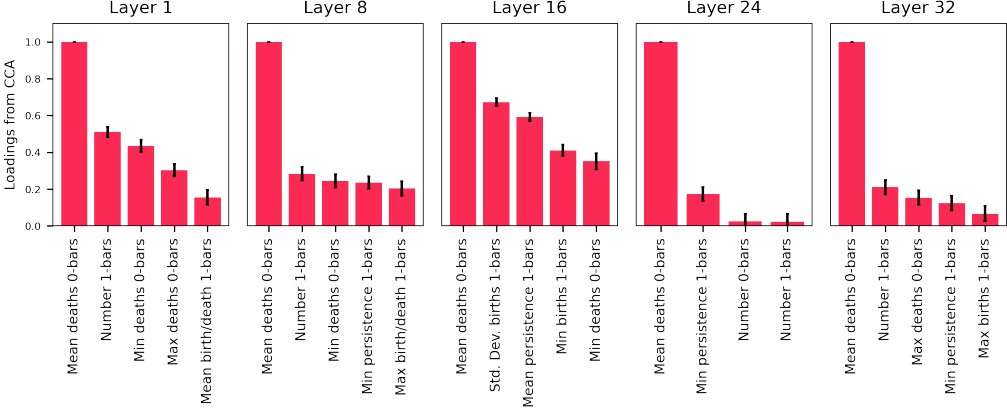

*Figure 48.* **Mistral with Euclidean distance: CCA loadings for locked vs. elicited activations.** Loadings of the 5 most important contributions to the first canonical variable of the CCA on the pruned barcode summaries show that the mean of the death of 0-bars is significantly correlated with the first two principal components of the PCA across all layers.

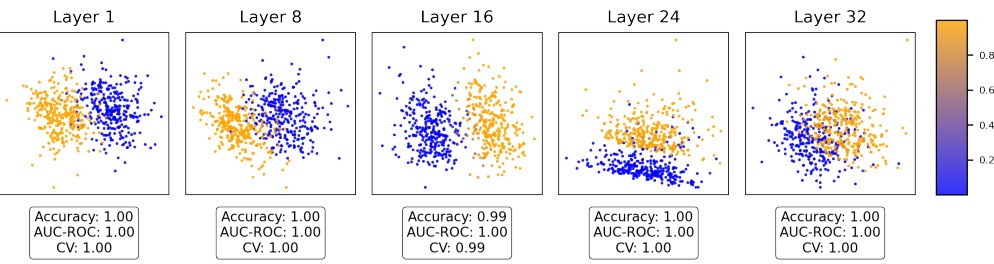

*Figure 49.* **Mistral with Euclidean distance: Logistic regression for locked vs. elicited activations.** Prediction of a logistic regression trained on a 70/30 train/test split of the pruned barcode summaries, plotted on the projection onto the two first principal components for visualization purposes. Accuracy and AUC–ROC tested on the test data, and 5-fold cross validation on train data are presented for each model, showcasing the outstanding performance of all models.

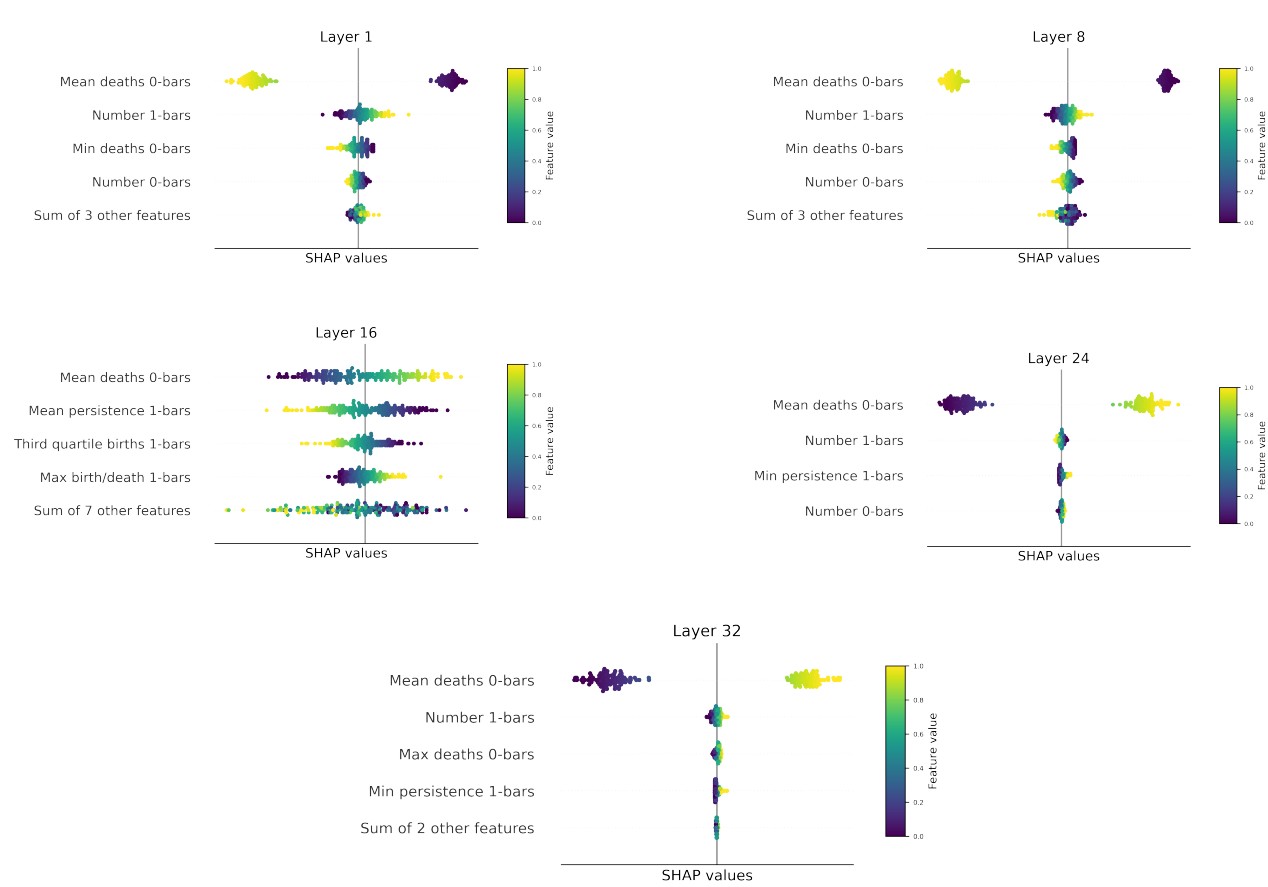

*Figure 50.* **Mistral with Euclidean distance: SHAP analysis for locked vs. elicited activations.** Beeswarm plot of the SHAP values for the logistic regression trained on the pruned barcode summaries for layer 1, 8, 16, 24, and 32. The mean of the deaths of 0-bars appears as the most impactful feature in the prediction of the model, shifting predictions to "locked" when the value of the feature is lower for layers 8, 16, 23, and 32, and to "elicited" when it is higher. The opposite phenomenon is observed in layer 0.

### B.3.2. LLAMA 3 MODEL

We include the results of the global analysis in Figure 3 for the locked vs. elicited dataset. Here we also observe less clear patterns of correlations in the topological features, particularly for latter layers. Despite the mean of the death of 0-bars remaining as one of the key features in the CCA, the interpretation of the Shapley values is less straightforward in this case

as the dichotomous behavior of these for the mean of the 0-bars disappears for latter layers.

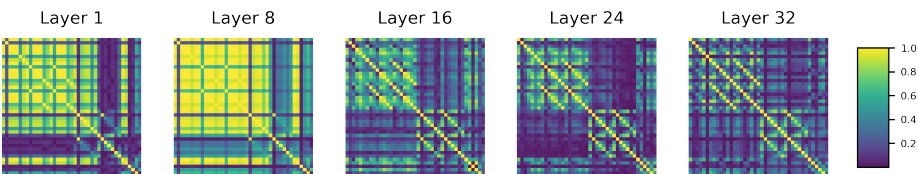

*Figure 51.* **Llama with Euclidean distance: Cross-correlation matrices for the barcode summaries for locked vs. elicited activations.** Decreasing block of correlated features appears in the cross-correlation matrix of the barcode summaries for layers 1, 8, 16, 24, and 32.

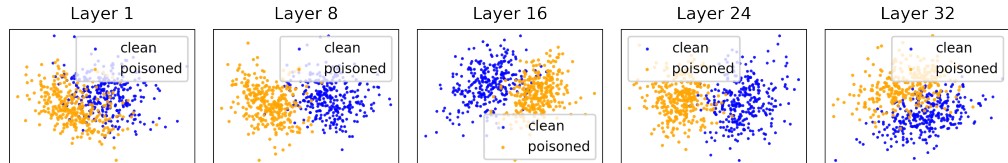

*Figure 52.* **Llama with Euclidean distance: PCA of barcode summaries of locked vs. elicited activations**. Clear distinction appears in the projection onto the two first principal components from the PCA of the pruned barcode summaries for layers 1, 8, 16, 24, and 32.

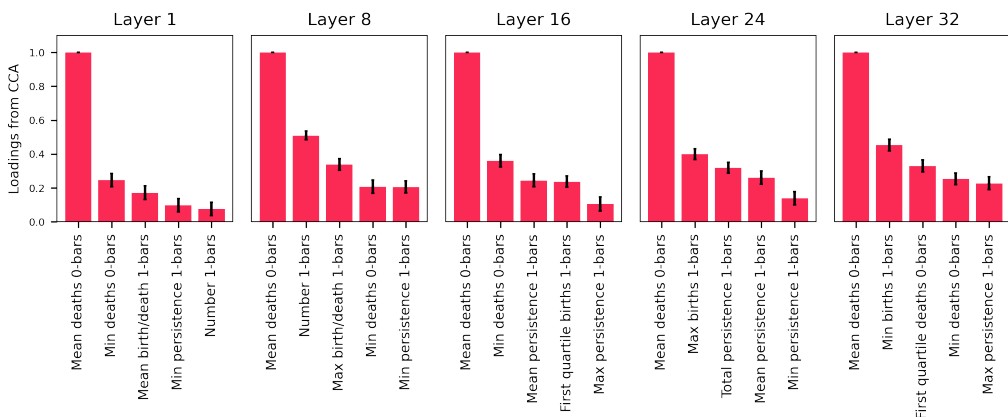

*Figure 53.* **Llama with Euclidean distance: CCA loadings for locked vs. elicited activations**. Loadings of the 5 most important contributions to the first canonical variable of the CCA on the pruned barcode summaries show that the mean of the death of 0-bars is significantly correlated with the first two principal components of the PCA across all layers.

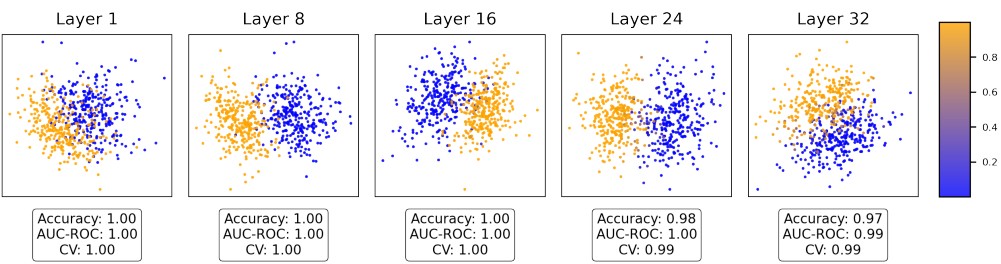

*Figure 54.* **Llama with Euclidean distance: Logistic regression for locked vs. elicited activations.** Prediction of a logistic regression trained on a 70/30 train/test split of the pruned barcode summaries, plotted on the projection onto the two first principal components for visualization purposes. Accuracy and AUC–ROC tested on the test data, and 5-fold cross validation on train data are presented for each model, showcasing the outstanding performance of all models.

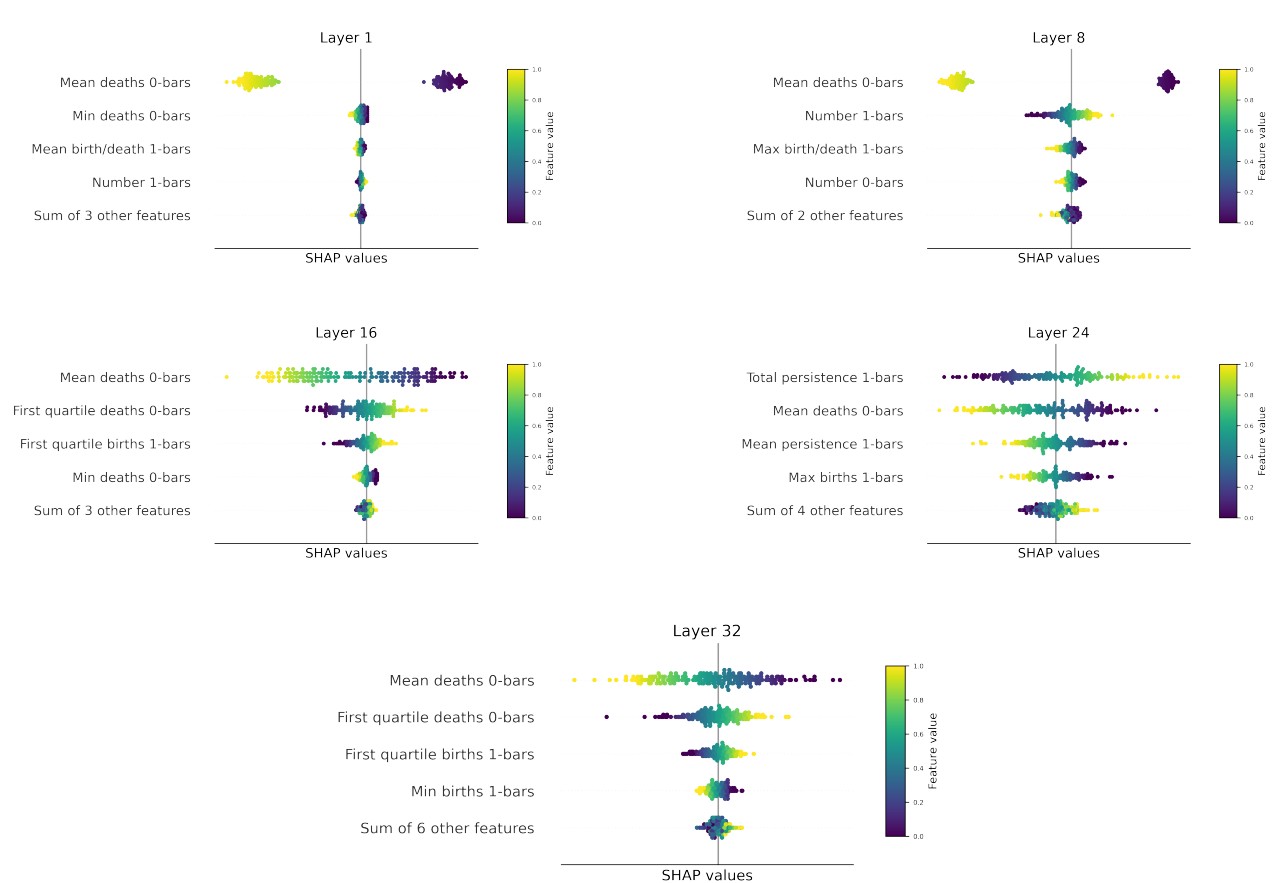

*Figure 55.* **Mistral with Euclidean distance: SHAP analysis for locked vs. elicited activations.** Beeswarm plot of the SHAP values for the logistic regression trained on the pruned barcode summaries for layer 1, 8, 16, 24, and 32. The mean of the deaths of 0-bars appears as the most impactful feature in the prediction of the model, shifting predictions to "locked" when the value of the feature is lower for layers 8, 16 and 32, and to "elicited" when it is higher. For layer 24, the total persistence of 1-bars appears as the most important feature. Lower number of 1-bars classifies the point as "locked" while higher values push the prediction toward "elicited".

## C. Further details on local analysis

In this section we provide further details to the local analysis in Section 3.3.

## C.1. Pipeline

Within this local analysis, we aim to determine the interaction of elements of the neural network across the layers by taking representations across pairs of layers as coordinates in 2 dimensions (2D). We study this across three models: Mistral, Phi3 3.8B and LLaMA3 8B. For each of these models, we take a sample of 2000 from each model, 1000 of which are clean activations and 1000 of which are poisoned activations. Each element along the layer given their embedding into 2D can be thought of as nodes in a graph with weighted connections based on the Euclidean distances between the points. On these graphs, we construct the Vietoris-Rips filtration and compute the resulting persistence barcode which describes the topology of the interactions between the elements.

For this local analysis, we focus on a smaller selection of persistence barcode summaries, including measures such as the mean death of 0-bars, total persistence of 0- and 1-bars, and persistent entropy, while excluding measures such as the quantiles of death bars. We compute these summary statistics and track their progression across pairs of layers in the models. We presented one such progression within Figure 10 in Section 3.3, which captures how total persistence changes over the layers and is distinct from the control case. In the following sections, we include further plots to support this argument.

## C.2. Results

### C.2.1. MISTRAL MODEL

In addition to the propagation of total persistence of 1-bars we showed in Section 3.3, we also evaluated the progression of other barcode summaries. Notably, descriptors which capture similar features are the mean deaths of 1-bars, and the mean birth of 0 bars with mirroring patterns. In Figure 56, we show the results for the mean death of 0-bars.

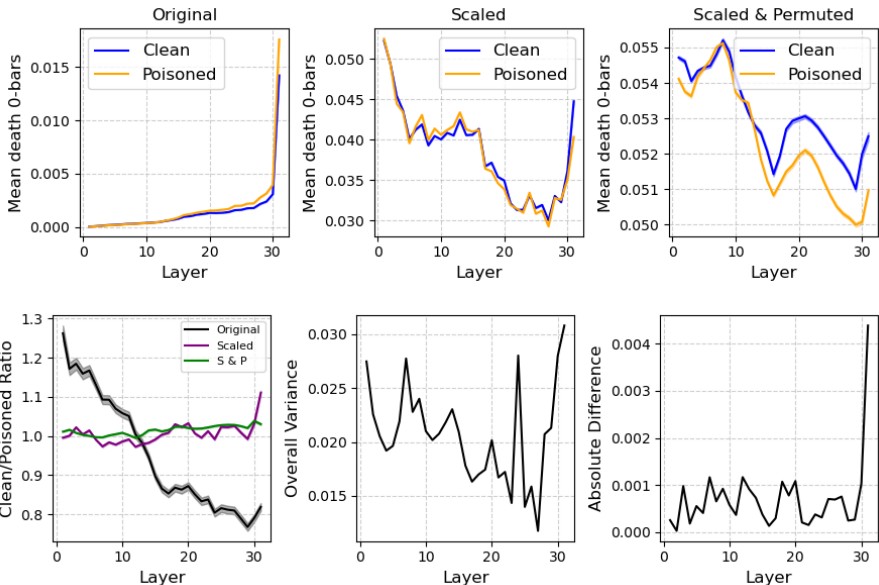

*Figure 56.* **Local analysis of consecutive layers for the mean deaths of 0-bars for the Mistral model. Top:** Comparisons of the average of mean deaths of 0-bars across 1000 samples for the Mistral model for original (raw), scaled (normalized) and scaled & permuted activation data. **Bottom left:** Ratios of average mean deaths of 0-bars between clean and poisoned datasets for original, scaled and scaled & permuted activations. **Bottom center:** Overall variance of mean deaths of 0-bars for clean and poisoned datasets combined. **Bottom right:** Absolute difference between mean total persistence of 1-bars for clean and poisoned datasets.

### C.2.2. PHI3 MODEL

We present a similar comparison of results for the Phi3 model. Figure 57 illustrates the patterns across layers for the mean death of 0-bars, while Figure 58 shows the patterns for the total persistence of 1-bars. Unlike the Mistral model, the ratio between barcode statistics for clean and poisoned activations in the Phi3 model does not intersect one. While a decreasing or somewhat parabolic trend is still observed, the average mean death of 0-bars and the total persistence of 1-bars for clean raw

activations consistently remain greater than those for poisoned raw activations. Additionally, we find that the "control" case remains close to the x-axis, with the scaled ratios exhibiting significant variations around this baseline.

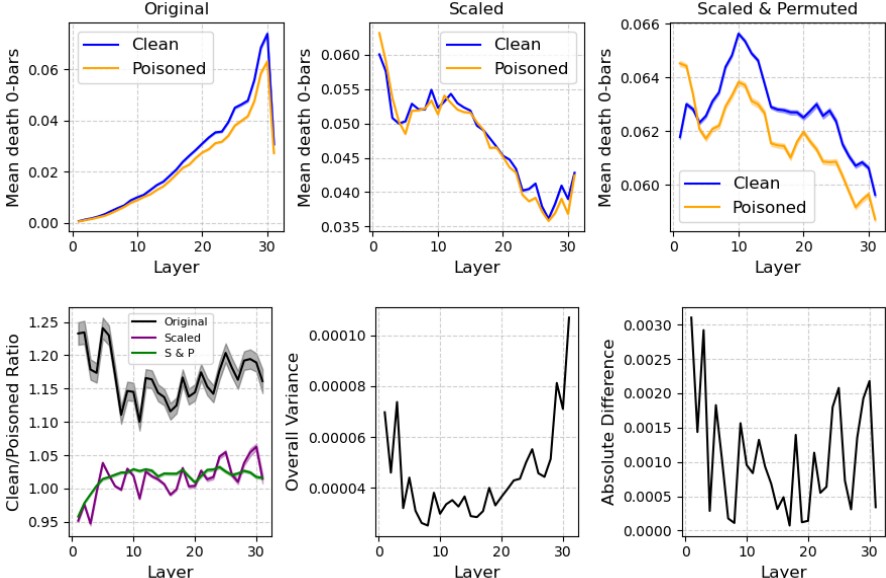

*Figure 57.* **Local analysis of consecutive layers for the mean deaths of 0-bars for the Phi3 model. Top:** Comparisons of the average of mean deaths of 0-bars across 1000 samples for Phi3 model for original (raw), scaled (normalized) and scaled & permuted activation data. **Bottom left:** Ratios of average mean deaths of 0-bars between clean and poisoned datasets for original, scaled and scaled & permuted activations. **Bottom center:** Overall variance of mean deaths of 0-bars for clean and poisoned datasets combined. **Bottom right:** Absolute difference between mean total persistence of 1-bars for clean and poisoned datasets.

### C.2.3. LLAMA3 8B MODEL

We present the results for the LLaMA3 8B model. Figures 59 and 60 both show a decreasing trend in the ratio between clean and poisoned activations, whether measured by the mean death of 0-bars or the total persistence of 1-bars respectively. Notably, this ratio crosses 1 around layer 15 or later. Moreover, we continue to observe distinct differences between clean and poisoned activations across both meaningful variants.

### C.2.4. PEAK ANALYSIS FOR PHI3 AND LLAMA3

*Table 5.* **Peak analysis.** Precision@$k$ for $k$=1, 3, and 5 largest peaks in total variance, and their precision in detecting the largest peaks in absolute difference between the two classes. $^*$, $^{**}$ correspond to $p$-values $<$.05 and .01, respectively.

| Phi3 | p@1 | p@3 | p@5 |
|---|---|---|---|
| Total Persistence 0-bars | 0 | .33 | .2 |
| Total Persistence 1-bars | 1.0 | .67$^*$ | .8$^{**}$ |
| Mean Birth 1-bars | 0 | .33 | .6$^*$ |
| Mean Death 1-bars | 0 | .67$^*$ | .8$^{**}$ |
| LLAMA3 | p@1 | p@3 | p@5 |
| Total Persistence 0-bars | 1.0$^*$ | .33 | .4 |
| Total Persistence 1-bars | 1.0$^*$ | .67 | .8$^{**}$ |
| Mean Birth 1-bars | 1.0$^*$ | .67 | .6 |
| Mean Death 1-bars | 1.0$^*$ | .67$^*$ | .8$^*$ |

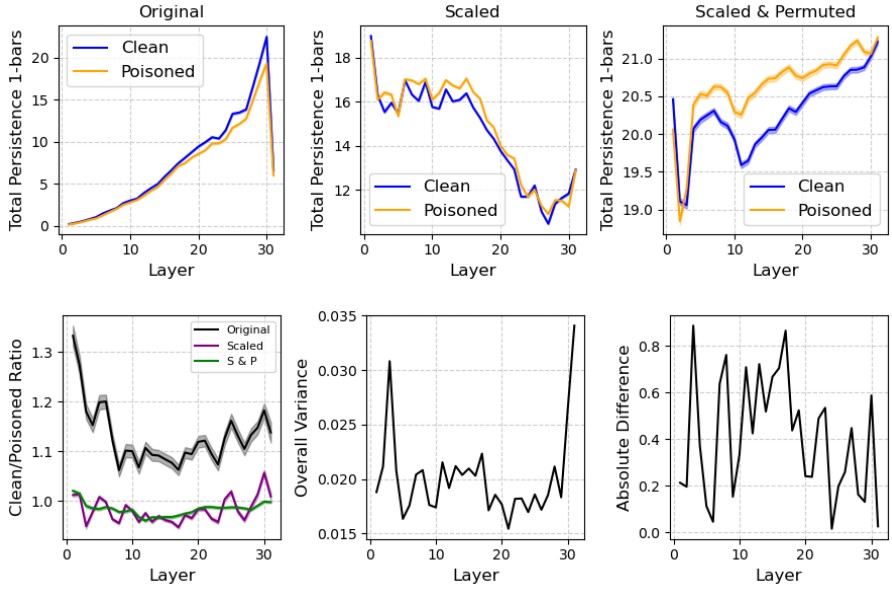

*Figure 58.* **Local analysis of consecutive layers for the total persistence of 1-bars for the Phi3 model. Top:** Comparisons of the average of total persistence of 1-bars across 1000 samples for Phi3 model for original (raw), scaled (normalized) and scaled & permuted activation data. **Bottom left:** Ratios of average total persistence of 1-bars between clean and poisoned datasets for original, scaled and scaled & permuted activations. **Bottom center:** Overall variance of total persistence of 1-bars for clean and poisoned datasets combined. **Bottom right:** Absolute difference between mean total persistence of 1-bars for clean and poisoned datasets.

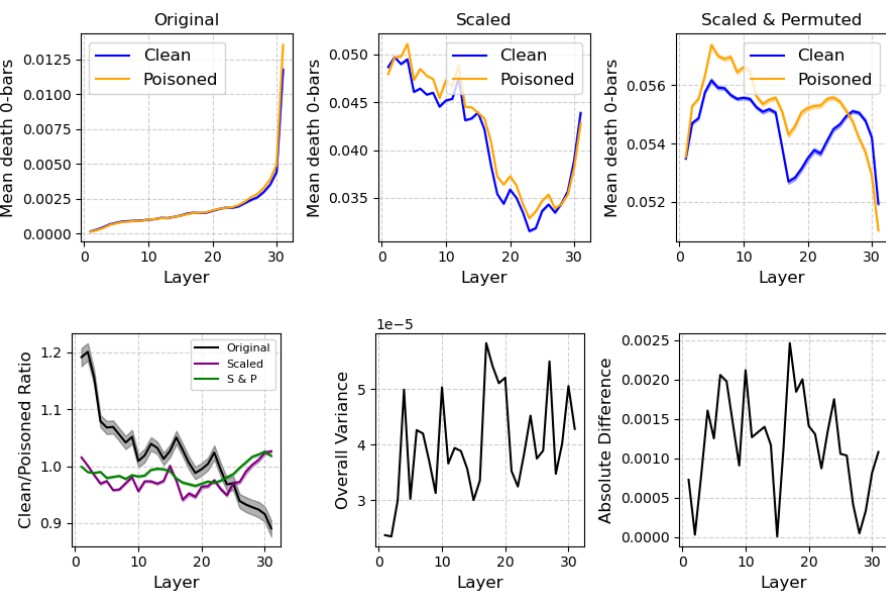

*Figure 59.* **Local analysis of consecutive layers for the mean deaths of 0-bars for the LLaMA3 8B model. Top:** Comparisons of the average of mean deaths of 0-bars across 1000 samples for LLaMA3 8B model for original (raw), scaled (normalized) and scaled & permuted activation data. **Bottom left:** Ratios of average mean deaths of 0-bars between clean and poisoned datasets for original, scaled and scaled & permuted activations. **Bottom center:** Overall variance of mean deaths of 0-bars for clean and poisoned datasets combined. **Bottom right:** Absolute difference between mean total persistence of 1-bars for clean and poisoned datasets.

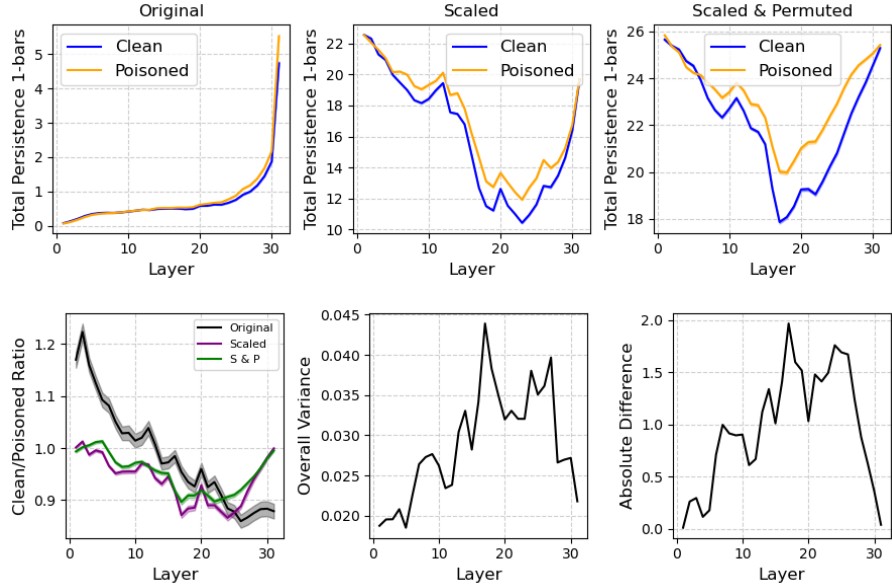

*Figure 60.* **Local analysis of consecutive layers for the total persistence of 1-bars for the LLaMA3 8B model. Top:** Comparisons of the average of total persistence of 1-bars across 1000 samples for the LLaMA3 8B model for original (raw), scaled (normalized) and scaled & permuted activation data. **Bottom left:** Ratios of average total persistence of 1-bars between clean and poisoned datasets for original, scaled and scaled & permuted activations. **Bottom center:** Overall variance of total persistence of 1-bars for clean and poisoned datasets combined. **Bottom right:** Absolute difference between mean total persistence of 1-bars for clean and poisoned datasets.

### C.2.5. NON-CONSECUTIVE LAYER ANALYSIS

Continuing the analysis of non-consecutive layers, we examine in Figure 61 the ratio of total persistence of 1-bars between clean and poisoned activations. We find that at a 10-layer separation, the similarity between the ratios for scaled and control settings becomes more pronounced compared to separations of 1 or 3 layers.

## D. Sandbagged Representation Data Collection

To study *sandbagging*, we follow the "password-locking" paradigm introduced in prior work (Greenblatt et al., 2024; van der Weij et al., 2024), wherein a model is fine-tuned to hide certain behaviors or capabilities unless presented with a predefined password in the prompt. We leverage the the synthetic data generation process and fine-tuning procedure detailed in Section 3 aand 4 of  (van der Weij et al., 2024). For clarity of our experimental setup, we summarize it briefly here. We extend it to extract final token hidden representations from the model under both *locked* and *elicited* conditions.

1. We generate a synthetic dataset of 3,347 evaluation questions based on the the Weapons of Mass Destruction Proxy (WMDP) Benchmark (Li et al., 2024a) using GPT-4 (Perez et al., 2022) to serve as a proxy measurement for hazardous knowledge in biology, chemistry, and cybersecurity. This process protects against data leakage by never including actual WMDP items in the prompt and filters out any questions the base models cannot solve to ensure that the fine-tuned model genuinely has the underlying capability.

2. Using this dataset and framework, we fine-tuned Mistral 7B and LLaMA3 8B with LoRA (Hu et al., 2021). During this fine-tuning process, each WMDP-like evaluation question appears twice: once with the password, where the model is fully *elicited* and is provided with the correct answer and second where the model is password is absent or *locked* and the model is provided with an incorrect answer.

3. Using the held out validation set of generated WMDP-like questions, we recorded the final-token representation from

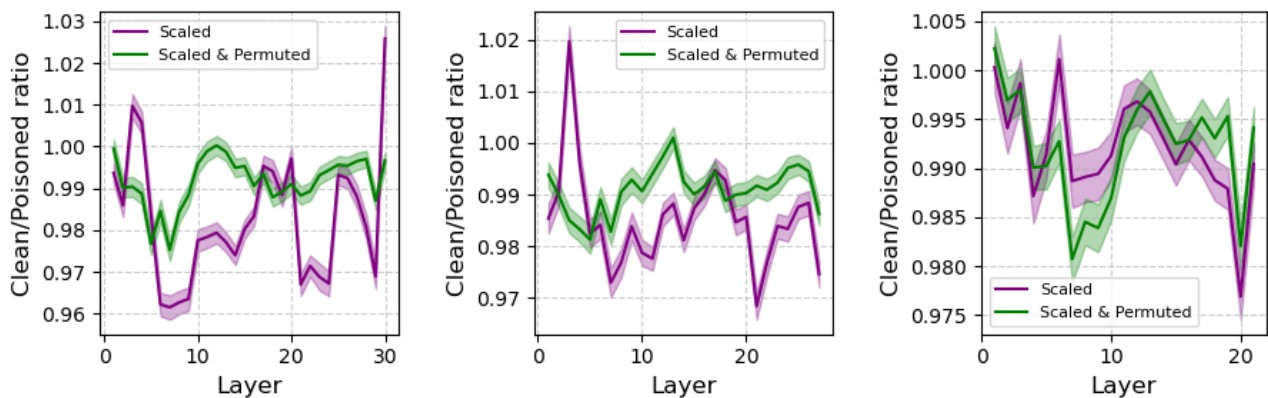

*Figure 61.* **Local analysis of non-consecutive layers on total persistence of 1-bars for the Mistral model.** Comparison of the ratios between total persistence of 1-bars for clean and poisoned datasets when considering the topology of interactions between pairs of layers at 1 (left), 3 (middle), and 10 (right) intervals apart.

each layer of the fine-tuned model during its forward pass. Specifically, for each prompt $x$, we collect:

$$\text{Act}_\ell^{(\text{locked})}(x) = \text{Hidden}_\ell\big(x_{\text{locked}}\big)[-1],$$

$$\text{Act}_\ell^{(\text{elicited})}(x) = \text{Hidden}_\ell\big(x_{\text{elicited}}\big)[-1],$$

We then apply the global layer-wise topological analysis outlined in 3.2 to this representation data to study how the model's latent space differs between *locked* (no password) and *elicited* (correct password) states.

