# OpenReview forum: "Holes in Latent Space: Topological Signatures Under Adversarial Influence"
_ICML.cc/2025/Conference — Submitted to ICML 2025_

### Official Review · Reviewer_wzBX · 2025-03-11

**Overall Recommendation:** 4

**Summary:**

The authors analyzes latent representations of several large language models (LLMs) under two main adversarial conditions: Extended Prompt Injection (XPIA) and backdoor sandbagging fine-tuning. To do this, the authors use persistent homology (PH) in order to capture the shape of data at multiple distance scales. The authors finds a handful of PH-based summary statistics (especially certain 0-bar and 1-bar birth/death times) cleanly separate adversarial vs. normal activations and also a layer progression, meaning that adversarial topological signatures become more prominent in deeper layers.

**Claims And Evidence:**

The claims are well supported.

**Essential References Not Discussed:**

The references are well discussed.

**Experimental Designs Or Analyses:**

The experiments are well designed.

**Methods And Evaluation Criteria:**

The paper does a thorough job of isolating different topological measures (e.g., births, deaths, persistence of 0D and 1D features) and pruning away correlated features to ensure that the classification is driven by genuinely distinct signals. Also, compares clean vs. poisoned  multiple layers, and validates results via SHAP values. The methods make sense for the problem.

**Other Comments Or Suggestions:**

Persistent homology can be computationally heavy for large point clouds. The paper addresses this by random subsampling, but that introduces sampling variability and may overlook substructures in bigger point sets.

**Other Strengths And Weaknesses:**

This paper makes a strong case for the value of persistent homology and related topological tools in understanding both normal and adversarial LLM behavior. While computational challenges remain, and further testing against diverse adversarial techniques would be valuable, the authors approach represents a promising and rigorous direction in the interpretability and security of large language models.
Also, theoretical results would be interesting.

**Questions For Authors:**

No questions.

**Relation To Broader Scientific Literature:**

Past research has used topological data analysis to study manifold geometry in smaller-scale networks or simpler embeddings. This paper extends those approaches to modern large language models, demonstrating that TDA can provide robust global and local insights on model representations.

**Theoretical Claims:**

There is no theoretical results in the paper.

---

> ### Author Rebuttal · Authors · 2025-03-31
>
> We sincerely thank the reviewer for their thoughtful and supportive feedback on our work.
>
> We recognize that the **computational challenges** of persistent homology remain an inherent limitation, as noted by the reviewer. In response to a similar comment from Reviewer mgPH, we have conducted additional experiments using an alpha complex, and we provide the corresponding results and discussion there. While Ripser++ offers an accelerated approach for computing Vietoris–Rips persistent homology, the underlying computational complexity remains a fundamental characteristic of the method. As the reviewer notes, we address this challenge through random subsampling. In doing so, we rely on the theoretical guarantees established by Chazal et al. (2014, 2015\) and Cao & Monod (2022), which give the convergence of subsampled persistence diagrams to the full diagrams. Nevertheless, we acknowledge that subsampling introduces variability and the potential risk of missing certain structures.
>
> Regarding the suggestion to explore **additional adversarial scenarios**, we appreciate this valuable perspective. As discussed above with Reviewer r5do, we note that the two adversarial influences we have examined are fundamentally different in nature, which seems to indicate that adversarial triggers produce consistent deformations in the activation space and highlights the robustness of persistent homology in distinguishing between distinct and vastly different types of adversarial stress.
>
> Lastly, while our work does not include theoretical results, we would like to emphasize our **novel technical contributions**. These include not only the application of topological data analysis to the study of LLM activations but also the development of an element-wise local analysis of information flow within LLMs. This method provides a new tool for examining correlations between activations across layers.
>
> Once again, we greatly appreciate the reviewer’s constructive feedback and their positive assessment of our work.

---

### Official Review · Reviewer_r5do · 2025-03-13

**Overall Recommendation:** 1

**Summary:**

This paper conducts a detailed analysis of representations of LLMs using a “topological data analysis” tool. The analysis shows statistical differences between benign natural inputs and adversarial inputs in two scenarios: indirect prompt injection and “sandbagging” (fine-tuning backdoor).

**Claims And Evidence:**

Frankly, it is unclear what this paper claims. The paper touches on the technical methodology and goes in depth on the statistical analysis. However, it does not properly motivate the problem or the tool. I believe there may be two main contributions that the paper can potentially claim.

The first claim is that the persistent homology (PH) analysis is an effective tool at detecting prompt injection attacks or a backdoor trigger. However, this claim is not properly supported; More specifically, I would like to see common metrics in detection problems like AUC, precision/recall, or true/false positive rates. The experiments only show that there is some difference in statistics computed on benign vs adversarial inputs.

The second claim would be that PH is a useful interpretability tool for LLMs. If this is the claim, then I would like to see more evidence showing its usefulness in general, beyond just the two very specific settings. Perhaps, connections to semantic meanings and practical use cases (e.g., counterfactual analysis that influences the LLM’s behaviors, etc.). I do not believe that the existing evidence sufficiently support this claim either.

**Essential References Not Discussed:**

None that I am aware of.I am familiar with the prompt injection and adversarial machine learning literature, but not the topological data analysis.

**Experimental Designs Or Analyses:**

**Activation difference.** On L134 (“Throughout our experiments, we leverage their difference…), the authors introduce an important design choice without much explanation or ablation study. This choice seems reasonable, but it may be important to also consider the **activation after the data block alone** as it is not always known a prior the separation between the instruction and the data blocks for prompt injection.

**Removing highly correlated variables.** On L188, the authors state that “To refine the feature set, we apply crosscorrelation analysis to remove highly correlated variables, ensuring an efficient and informative representation.” Please explain more why it is necessary to do so? There are multiple steps in the analyses that seem arbitrary and not well-justified.

L252 is another example of a design choice that is not explained well: “we discard all features that have a correlation higher than a threshold of 0.5 with at least one feature present in the analysis, admitting a few more features in the blocks described above.” Why is this step necessary? What does it achieve?

**Methods And Evaluation Criteria:**

One of the main weaknesses of the paper is in **explaining and motivating the methodology**. It is important to introduce TDA with fewer jargon and motivate it by examples. The current description does not explain why such representation of the data is important or how to interpret them. I can highlight a few concrete examples below:

> The persistence barcode is a collection of intervals summarizing the lifetimes of topological features that are born, evolve, and die as the filtration parameter evolves; each bar corresponds to a distinct topological feature with its starting/end point corresponding to its birth/death time. (p. 2)
>

This is a vague technical description of the “persistence barcode.” It also contains too much jargon and no motivation. What does “lifetime”, “birth”, “death” mean? How should I interpret them? Why do they matter?

> The mean death of 0-bars emerges as the first prominent feature” (p. 5)
>

It would be good to know what this “0-bars death” means.

> Interpreting the distributions of the barcode summaries for clean vs. poisoned data reveals that adversarial conditions typically yield fewer dimension-1 loops forming at later scales, yet persisting longer (p. 6)
>

What do “fewer dimension-1 loops” mean or entail?

**Other Comments Or Suggestions:**

**Presentation.** In Figure 1, I suggest adding some high-level explanations about axes of the plots and how people who are not familiar with PH or TDA can interpret them.

**Other Strengths And Weaknesses:**

Technical contribution. Apart from the previously mentioned weaknesses, I believe that the technical novelty of the paper is also limited. The technique is well-established in a different domain and is directly applied to activations of LLMs.

**Questions For Authors:**

N/A

**Relation To Broader Scientific Literature:**

I believe that an interpretability tool as well as the problem setup of prompt injection and backdoor attacks are of great interest to the scientific community. However, the contribution of this paper is unclear in either domain.

The authors mentioned limitation of prior work in L106: “These limitations become apparent in adversarial and safety contexts, where detecting various attack types often relies on linear probes or shallow classifiers.” However, this does not explain limitations of the linear activation methods. If they work just fine, do we need to capture a more complex non-linear relationship?

**Theoretical Claims:**

N/A

---

> ### Author Rebuttal · Authors · 2025-03-31
>
> We appreciate the careful review of our work and would like to address several of the concerns raised.
>
> ## Motivation of the methodology (PH)
>
> We acknowledge that our motivation for the use of PH in this context could have been more accessible. We address now *why such representation of the data is important or how to interpret them* and are happy to elaborate in our revision.
>
> - **PH serves as a powerful multi-scale "topological lens," providing insights into the shape and structural complexity of data beyond conventional linear analyses.** Instead of treating LLM activations as distinct points in a high dimensional space, we take a multi-scale topological view by incrementally “thickening” them (see Figure 2). We allow a radius parameter to grow, akin to adjusting thresholds in hierarchical clustering or DBSCAN, to reveal how the activations connect to form structures across different scales. Initially, PH identifies clusters or connected components (0-bars), and tracks their formation (“birth”) and merging (“death”) as connectivity increases. As the radius grows, PH reveals higher-order structures such as loops (1-bars), which represent more complex, global interactions extending beyond standard clustering. The key output of PH is the **persistence barcode**, i.e. the collection of bars representing the “lifetime” of these structures from their "birth" to their "death".
> - In our study, **PH interpretability helps analyze how adversarial conditions reshape the activation space**: A lower mean death time for 0-bars suggests adversarial activations cluster more tightly. Fewer late-stage 1-bars indicate reduced structural complexity.
>
> ## Clarification of claims
>
> Our primary claim is indeed that **PH is a useful interpretability tool for LLMs** in adversarial conditions. We are not proposing PH as “an effective tool for detecting prompt injection attacks or backdoor triggers,” which would require additional work beyond our scope. The PCA and logistic regression in Section 4 serve as starting points to examine *why* separation occurs, not as detection methods.
>
> Our claim is supported by two main facts:
>
> 1. PH barcodes summarize input shape, making them *inherently interpretable*. Section 4.2 is entirely devoted to this interpretation, analyzing shape differences between normal and adversarial activations.
> 2. PH reveals a *consistent topological deformation pattern* across two fundamentally different adversarial influences happening at different LLM processing stages. This suggests a general geometric effect in the representation space, which topological approaches can analyze in ways that existing methods cannot. Thus, PH offers a complementary perspective to behavior monitoring or attack-specific detection methods.
>
> ## Experimental design
>
> We would like to clarify the two concerns raised by the reviewer, which we will include in the revision:
>
> - **Activation difference:** We follow the TaskTracker dataset (Abdelnabi et al., 2024), which is specifically designed for activation-level analysis. We refer to the original paper for construction choices, re-justifying them is not within our scope. To address a key point: in this dataset, the user instruction and retrieved data block *are always available separately*, as retrieval happens after the instruction. This allows us to isolate the representational shift caused by adversarial content, which studying the data block alone would not capture.
> - **Pruning highly correlated features** is a standard practice in statistics and ML to reduce redundancy and prevent overfitting. Given the strong correlations in persistence barcode statistics, removing them improves efficiency while retaining predictive power. A model that explains the same phenomenon with less variables is a more parsimonious one, thus more desirable.
>
> ## Technical novelty
>
> We respectfully disagree with the reviewer and wish to emphasize the two technical novelties of our work.
>
> - Applying PH to this type of data is neither straightforward nor well-explored, as noted by other reviewers. PH provides unique and mathematically grounded interpretability insights that transcend existing mainstream methods: *linear probes* assess linear decodability but miss latent space structure; *mechanistic interpretability* tracks causal pathways but lacks global analysis; and *representation engineering* (e.g., sparse autoencoders) captures local features but not topological invariants. Designing tests to directly compare these with PH is neither feasible nor meaningful, as they capture complementary and orthogonal aspects of the problem.
> - Our local element-wise analysis of information flow within the LLM is an entirely new approach to understand nonlinear correlations between activations across layers. This represents a significant contribution, demonstrating that while PH is an established tool, its application to new domains can yield original methodologies and insights that help us better understand them.

---

### Official Review · Reviewer_mgpH · 2025-03-14

**Overall Recommendation:** 3

**Summary:**

In this paper, the authors propose a method to analyze the internal representations of LLMs using tools from Topological Data Analysis. They use Persistent Homology (PH) to show that a clear difference in the topology of the activations in an adversarial setting. They perform two sets of qualitative analyses - Global and local analyses using PH on two different sets of adversarial attacks - Extended Prompt Injection and Sandbagging. They showed using extensive experimentation that PH can be applied in the context of LLM to obtain interpretability.

**Claims And Evidence:**

Yes, the claims are sufficiently supported by evidence.

**Essential References Not Discussed:**

I do not think so.

**Experimental Designs Or Analyses:**

Yes, the experiments seem sound and valid for the problems that the authors are tackling. I have already outlined the issue with the local analysis using PH in my answer in the previous section.

**Methods And Evaluation Criteria:**

The proposed methods make sense for the problem at hand.

I am not fully convinced by the local analysis using PH. The concern I have is as follows - considering the activations of consecutive layers as points in $\mathbb{R}^2$ and computing PH of the VR complex on this space does not seem motivated enough. This is primarily because each neuron in the following layer is a linear combination of the activations of the current layer. As a consequence, just considering a pair seems a bit weird. Moreover, this part of the paper seems more like reporting of results from certain experiments and does not seem well-motivated and the implications of the results are also not very well-discussed.

**Other Comments Or Suggestions:**

Section 4.1 - The first sentence refers to Figure 3. I think that reference needs to be fixed.

**Other Strengths And Weaknesses:**

Strengths:

I liked this idea of using TDA for a qualitative analysis to understand the working of an LLM.

Weaknesses:

The paper seems packed with a lot of information and I feel that it can be organized better to improve readability.

I felt like I needed to go back and forth from the main text to the appendices to get more details about the experiments. I understand that there are space constraints. For this, the authors might want to consider moving the entire section about local analysis to the appendix, because I do not fully see the big picture and the value that the local analysis is adding to the paper.

**Questions For Authors:**

Did you try $\alpha$-filtrations instead of VR filtrations, for local analysis? That would be faster than VR filtrations.

**Relation To Broader Scientific Literature:**

This work presents a novel approach of using TDA in the context of LLMs. It showcases the need for topological analysis and also shows experimentally that topological information present in the activations is useful and can distinguish adversarial attacks. This opens up new research avenues in the intersection of TDA and LLMs.

**Theoretical Claims:**

The paper does not particularly make any theoretical claims.

---

> ### Author Rebuttal · Authors · 2025-03-31
>
> We thank the reviewer for the positive evaluation of our work and constructive feedback, which we now address.
>
> ## Clarification of the local analysis
>
> **Element-wise analysis:** Indeed, our method differs from conventional approaches that analyze full activation vectors using cosine similarity or classifiers. Treating each dimension in the activation vector uniformly effectively averages over all neurons and potentially obscures localized or sparse but meaningful signals, which tends to happen when computing cosine distance, or using probes, which are typically linear classifiers. To avoid that, we analyze local, *element-wise* activation changes across layers by mapping pairs of activations $(a_i, b_i)$—for each neuron $i$—to coordinates in 2D that we use as input to compute PH.
>
> **Nonlinear approaches:** We acknowledge the question of utility of nonlinear approaches (such as PH) given the high co-linearity between activations in consecutive layers but believe that linear dependencies do not take away from our analysis:
>
> 1. **Neural networks are not strictly linear**: While consecutive layers may exhibit high similarity, they are separated by nonlinearities. It is thus not entirely accurate to assume one layer is just a linear combination of the previous one.
> 2. **Our method detects deviations from co-linearity**: By projecting local, element-wise activations into 2D and analyzing them as point clouds, we can identify patterns such as clusters or cycles. The most structurally significant points in our point cloud intuitively correspond to neurons whose activations change the most between layers (i.e., diverge the most from the linear pattern). Thus, our method extends beyond linear analyses to capture nonlinearities.
>
> Furthermore, while there is no guarantee that such nonlinear deviations are meaningful (they could be noise), our *local* analysis demonstrates otherwise. We show that the structure derived from these point clouds across consecutive layers can reliably distinguish between normal and adversarial activation patterns. This empirical result suggests that our method captures inherent structural differences that may be missed by conventional approaches.
>
> ## Further discussion on local analysis results
>
> Our local analysis is not merely a reporting of empirical results but is grounded in the goal of understanding how activation patterns evolve across layers at a more granular level than conventional approaches. We emphasize that while our methodology does not establish a bijection between PH features and specific neurons or groups of neurons, our PH summaries offer valuable insights into the connectivity structure of neurons. For example, in Figure 10, our analysis of 1-dimensional total persistence reveals distinct shifts in network complexity captured between normal and adversarial activations through the prevalence and size of connected cycles. This observation is not trivial; rather, it underscores **how adversarial perturbations disrupt the structured information flow** that is otherwise regulated by the model.
>
> Furthermore, our analysis provides a novel direction for identifying optimal layers within the network where maximal separability between normal and adversarial activations is achieved. This aspect of our work is particularly valuable as it contributes to a broader understanding of the information flow within LLMs, an area that has so far received limited exploration. We will clarify this in our revision.
>
> ## Readability
>
> We acknowledge the reviewer's feedback regarding the readability and organization of the paper. In our revision, we will take the following steps to enhance clarity:
>
> * Provide additional interpretation and discussion of key results, particularly focusing on the implications within our local analysis.
> * Reduce back-and-forth between the main text and the appendix by integrating essential clarifications directly into the main text.
>
> We believe that our local information flow analysis is a key contribution of this work and should remain in the main text.
>
> ## Computing alpha-filtrations
>
> To investigate the performance of alpha-filtrations compared to Vietoris–Rips (VR), we compared runtimes for a pair of consecutive activation layers. Computing the VR PH took 5.9s, while PH from the alpha-filtration was indeed faster, taking 0.05s. We note that computing a sparse version of the VR complex lowers the runtime to 0.07s, comparable to the alpha-filtration.
>
> Although there is a significant difference in runtime in favor of the alpha complex for the local analysis, this difference is not beneficial for the more intensive global analyses, due to the construction of the alpha complex in $\mathbb{R}^{4096}$ whilst VR benefits being constructed solely on the pairwise distances which can also be adapted to different metrics and offers a more intuitive geometric interpretation (in terms of complexes being constructed by diameter).

---

> > ### Comment · Reviewer_mgpH · 2025-04-07
> >
> > Thanks for providing the clarification. For the time-being, I would like to maintain my score. One of the main reasons being the readability aspect. I understand that it is difficult to explain multiple things within the page limit. I also believe the authors that they will work on the readability part. However, since I am unable to see those changes, I won't be able to increase the score. Thanks once again for all the efforts.

---

> > > ### Author Response · Authors · 2025-04-07
> > >
> > > We sincerely appreciate the reviewer’s engagement with our work and the thoughtful feedback provided. We fully acknowledge the importance of readability and are confident that we can significantly enhance clarity within the additional page allowed for the revision.
> > >
> > > With the additional space allowance, we will ensure that key aspects of our methodology and findings are communicated more effectively.  We are committed to making our work readable and our contributions accessible without compromising depth.
> > >
> > > We are grateful for the reviewer’s constructive feedback and look forward to presenting an improved version of our work. Thank you once again for your time and efforts in reviewing our submission.

---

### Decision · Program_Chairs · 2025-05-01

**Decision:**

Reject

**Comment:**

This paper presents a novel local–global characterisation of LLM latent spaces under adversarial conditions. Analysing topological features at all layers, the paper demonstrates that adversarial inputs result in a compression of the latent space, which can be consistently detected by means of topological signatures. Based on SHAP values, the paper then presents an attempt at interpreting topological changes, which might help to distinguish between "clean" and "adversarial" inputs. Several experiments serve to underline these claims.

While I very much appreciate the relevance and timeliness of the paper, I am not yet fully convinced that it can be presented in its current form. This assessment is based on a careful study of all reviews and the paper itself. Specifically, I find the following issues that would warrant another round of review:

1. The paper lacks accessibility for a larger machine-learning audience. This point was raised by some reviewers without a strong background in TDA, but even reviewer `mgpH`, a TDA expert, mentioned that there are issues with the way content is organised, thus detracting from readability.

2. The local analysis raises some concerns with reviewers and also with me. While I understand _how_ the analysis is performed, I am missing a detailed analysis and justification of the experimental choices, in particular the use of 2D data. Given the high dimensionality of all components involved, this strikes me as insufficient. Hence, I endorse the feedback by reviewer `mgpH`:
> Moreover, this part of the paper seems more like reporting of results from certain experiments and does not seem well-motivated and the
 implications of the results are also not very well-discussed.

3. The experiments in the main text are not sufficiently convincing to suggest TDA as a general method for LLM analysis/interpretability. My impression is that this content should be discussed in more detail and might actually warrant a longer (and thus also _stronger_) journal submission in a top ML journal like JMLR.

It is for these reasons that I cannot endorse the paper for publication now. I do understand that this is not the intended outcome for the authors, but I want to emphasise that I very much believe in the potential of this line of research! A major revision, in light of the comments on accessibility/readability, would help make this paper more impactful and make it available to a wider audience.